# Death by a thousand cuts through kinase inhibitor combinations that maximize selectivity and enable rational multitargeting

Ian R Outhwaite[1], Sukrit Singh[1,2], Benedict-Tilman Berger[3,4], Stefan Knapp[3,4], John D Chodera[2], Markus A Seeliger[1]*

[1]Department of Pharmacological Sciences, Stony Brook University, Stony Brook, United States; [2]Computational and Systems Biology Program, Sloan Kettering Institute, Memorial Sloan Kettering Cancer Center, New York, United States; [3]Institute of Pharmaceutical Chemistry, Goethe University Frankfurt, Frankfurt am Main, Germany; [4]Structural Genomics Consortium, Buchmann Institute for Life Sciences, Goethe University Frankfurt, Frankfurt am Main, Germany

*For correspondence:
markus.seeliger@stonybrook.edu

**Abstract** Kinase inhibitors are successful therapeutics in the treatment of cancers and autoimmune diseases and are useful tools in biomedical research. However, the high sequence and structural conservation of the catalytic kinase domain complicate the development of selective kinase inhibitors. Inhibition of off-target kinases makes it difficult to study the mechanism of inhibitors in biological systems. Current efforts focus on the development of inhibitors with improved selectivity. Here, we present an alternative solution to this problem by combining inhibitors with divergent off-target effects. We develop a multicompound–multitarget scoring (MMS) method that combines inhibitors to maximize target inhibition and to minimize off-target inhibition. Additionally, this framework enables optimization of inhibitor combinations for multiple on-targets. Using MMS with published kinase inhibitor datasets we determine potent inhibitor combinations for target kinases with better selectivity than the most selective single inhibitor and validate the predicted effect and selectivity of inhibitor combinations using in vitro and in cellulo techniques. MMS greatly enhances selectivity in rational multitargeting applications. The MMS framework is generalizable to other non-kinase biological targets where compound selectivity is a challenge and diverse compound libraries are available.

## Editor's evaluation

This study presents a valuable finding on a multi-compound-multitarget scoring (MMS) method that combines inhibitors to maximize target inhibition and to minimize off-target inhibition. The strategy may enable the optimization of inhibitor combinations for multiple on-targets. The evidence supporting the claims of the authors is solid. The work will be of interest to pharmacology scientists working in both academic and industrial sectors.

## Introduction

The off-target effects of pharmacologic compounds against unintended targets represent a major challenge in biomedical research. The off-target activity of compounds is difficult to appreciate without extensive study (*Cichońska et al., 2021*; *Santos et al., 2017*; *Wauson et al., 2013*; *Lin et al.,*

**Figure 1.** Inhibitor combinations can reduce off-target activity. (**A**) Selectivity of kinase inhibitors is limited. Fold activity is the PKIS2 activity of each PKIS2-645 inhibitor against the kinase target it is maximally active against versus the target it is second most active against when screened at 1 μM in the DiscoverX KINOMEscan competitive displacement assay. No inhibitor has even twofold activity for its primary target versus its secondary target. PKIS2-645 inhibitors with 100% activity against multiple targets were excluded from the analysis. (**B**) Illustrative inhibition profiles of three hypothetical kinase inhibitors with equal activity against a target of interest and different off-targets; a combination of the three inhibitors retains on-target inhibition while diluting off-target effects.

The online version of this article includes the following figure supplement(s) for figure 1:

**Figure supplement 1.** Compound activity is a function of potency and concentration.

2019; *Dahlin et al., 2017*) and may complicate interpretation of observed phenotypes in experimental systems (*Lin et al., 2019*; *Giuliano et al., 2018*; *Palve et al., 2022*; *Emmerich et al., 2021*). Human protein kinases are a major target for small molecule compounds but even relatively selective kinase inhibitors exhibit significant off-target activity in large screening campaigns (*Santos et al., 2017*; *Fabian et al., 2005*; *Karaman et al., 2008*; *Davis et al., 2011*; *Elkins et al., 2016*; *Klaeger et al., 2017*; *Drewry et al., 2017*; *Deibler et al., 2017*; *Drewry et al., 2019*; *Wells et al., 2021*; *Figure 1a*). The off-target activity of kinase inhibitors can yield misleading results. For example, the mechanism of clinically evaluated anti-cancer compound OTS167 was later found to be due to CDK11 inhibition rather than the previously mischaracterized target MELK (*Lin et al., 2019*; *Giuliano et al., 2018*). Therefore, improving selectivity through minimizing inhibitor off-target activity is an important prerequisite for studying human protein kinases with kinase inhibitors in biological systems.

Obtaining selective single compounds is limited by both identification of off-targets and the challenges of modifying compounds to reduce off-target effects. In addition to screening technologies (*Miduturu et al., 2011*; *Jacoby et al., 2015*; *Nieman et al., 2023*), promising computational approaches may help identify compound off-targets or strategies to target particular kinases (*Sydow et al., 2022*; *Cichońska et al., 2021*; *Zhang et al., 2023*). However, even after compound off-targets are identified, techniques to improve compound selectivity such as rigidification or ring closure may

not be generalizable across chemotypes or improve selectivity for structurally similar off-targets (*Assadieskandar et al., 2018*; *Wu et al., 2021*). Allosteric inhibitors designed for a particular kinase target or even kinase mutants can have excellent selectivity (*Jia et al., 2016*; *Schoepfer et al., 2018*; *Wrobleski et al., 2019*), but these compounds are hard to engineer and may still bind to multiple targets (*Laufkötter et al., 2022*; *Mingione et al., 2023*; *Attwood et al., 2021*). Given the difficulty of obtaining selective inhibition for a single kinase it is not surprising that selective inhibition of a set of kinases would be an even harder problem.

Rational multitargeting, the simultaneous inhibition of multiple targets, is a major objective in biomedical research and drug design (*Sivakumar et al., 2020*; *Hopkins, 2008*; *Xiong et al., 2021*). One goal is to enable rational polypharmacology, a simultaneous targeting of compensatory signaling pathways to potentiate drug action or inhibit mechanisms of drug resistance (*Hammam et al., 2017*; *Bahcall et al., 2022*; *Quereda et al., 2019*). However, targeting multiple kinases, either through combination therapy or nonselective compounds, often leads to inhibition of off-targets and toxicity (*Ringheim et al., 2021*; *Panagiotou et al., 2022*). Similarly, studying the effects of co-inhibiting multiple targets is difficult due to the off-target effects of compounds. A method to selectively inhibit multiple targets would be a valuable tool to understand mechanistic causes for synergistic toxicities and how to identify targets for polypharmacology.

The large chemical space of kinase inhibitors and the comprehensive off-target characterization (*Santos et al., 2017*; *Fabian et al., 2005*; *Karaman et al., 2008*; *Davis et al., 2011*; *Elkins et al., 2016*; *Klaeger et al., 2017*; *Drewry et al., 2017*; *Deibler et al., 2017*; *Drewry et al., 2019*; *Wells et al., 2021*) of these compounds suggests that it would be possible to engineer combinations of inhibitors with shared effects at a kinase of interest but different off-target effects. These inhibitor combinations would dilute off-target activity, and improve selectivity for the target kinase (*Figure 1b*). In some cases, combinations of inhibitors might have better selectivity than any available single compounds. These combinations could be designed for studying single or sets of multiple target kinases. This rational multitargeting could provide a flexible strategy building on the utility of the otherwise rigid selectivity profiles of single chemical probes.

Here, we propose a multicompound–multitarget scoring (MMS) method to calculate the selectivity of combinations of kinase inhibitors and identify the most selective combination of compounds to inhibit single or multiple target kinases. The off-target activity of combinations of compounds is calculated, the selectivity of these combinations is determined, and the concentrations of compounds in these combinations are optimized to further reduce off-target activity and maximize selectivity. We implement this approach for single and sets of multiple kinase targets using currently available chemogenomic datasets (*Karaman et al., 2008*; *Davis et al., 2011*; *Klaeger et al., 2017*; *Drewry et al., 2017*). We validate the predicted activity and selectivity of inhibitor combination using in vitro and in cellulo techniques. As predicted, we find that the benefit of combinations depends dramatically on the available inhibitor dataset to inform combinations. While we predict and find a modest specificity improvement for single kinases, we predict and confirm a more pronounced benefit for multiple kinase inhibition through inhibitor combinations. We expect that the utility of this method will improve as the number of characterized kinase inhibitors increases by providing larger specificity gains and benefitting more kinases. The MMS method is generalizable to other non-kinase compound–target interaction systems and may inform future strategies to reduce off-target effects.

## Results

### Data types and definitions

Numerous data types are used to quantify compound–target interactions (*Tang et al., 2018*). We define the activity of an inhibitor as its fractional target occupancy at a given concentration. For example, occupation of 50% of Abl kinase molecules with imatinib at 10 nM would correspond to an imatinib activity of 50% at 10 nM. Inhibitor activities are cumulative: a mixture of kinase inhibitors that occupy 98% of Abl and 37% of Src kinases would have activities of 98% and 37% for Abl and Src, respectively. This inhibitor activity is a measure of target engagement and is nonlinear with respect to $K_d$, $K_d^{app}$, $K_i$, or $EC_{50}$ values (*Figure 1—figure supplement 1*). For example, a compound with a $K_i$ of 1 µM would report 50% inhibitor activity at 1 µM, while a second compound with a $K_i$ of 0.1 µM would reach ~90% inhibitor activity at 1 µM concentration. It is advantageous to consider selectivity

from this activity-scale perspective as log-fold differences in potency may have minor effects on the total percent of occupied targets if sites are already nearly fully saturated or unfilled (*Figure 1—figure supplement 1*).

The activity of inhibitors or combinations of inhibitors against off-target kinases are communicated in some figures as probability density functions (pdfs) (e.g. Figure 3f, *Figure 4—figure supplement 1b*, Figure 5f, *Figure 5—figure supplement 1b*). These plots treat off-target kinase activity as a probability rather than a single value to account for error in the calculation of the activity of inhibitor combinations. The $K_i$, $K_d$, or $EC_{50}$ of a compound or mixture of compounds for a single off-target kinase is used to calculate off-target compound activity given the concentration of the compound(s) required to reach minimum threshold on-target activity for all target kinase(s). This minimum on-target kinase threshold is 90% activity unless otherwise indicated. This single calculated off-target activity value is set as the mean of a Gaussian with an area under the curve of 1. The cumulative pdf as communicated in figures is the sum of all off-target Gaussians for a particular inhibitor or inhibitor combination.

The term fold-error is used to describe the accuracy of predicted $K_d$ or $EC_{50}$ values for inhibitor combinations. Fold-error is the predicted or observed value (whichever is larger) divided by the other, such that fold-error is always greater than or equal to 1, where a fold-error of 1 represents a perfectly accurate prediction.

We focus our analyses on a subset of quality kinase inhibitor datasets: *Karaman et al., 2008*, *Davis et al., 2011*, PKIS2-645 (*Drewry et al., 2017*), and *Klaeger et al., 2017*. These datasets are further described in Materials and methods: Dataset sources. Drewry et al. tested 645 inhibitors at 1 µM against 406 human protein kinases in a competitive displacement assay for inclusion in the Protein Kinase Inhibitor Set 2 (PKIS2) (*Drewry et al., 2017*). We refer to this set of 645 inhibitors as PKIS2-645 inhibitors and percent displacement in that assay as PKIS2 activity.

## The MMS method

The MMS method calculates the most selective way of obtaining on-target activity for a target or set of targets and may nominate combinations of inhibitors rather than single inhibitors for maximal selectivity (*Figure 2*). The five subsections of this method description correspond to the five components of *Figure 2*. Additional description is provided in Materials and methods.

First, sufficiently potent inhibitors and combinations of inhibitors are identified. A threshold of 90% PKIS2 activity or a $K_i/K_d/K_d^{app}$ of 111 nM against the target is used to define an inhibitor as sufficiently potent.

Second, an off-target activity range of interest is selected. For example, high off-target effects greater than 50% activity may be important in a particular experimental context. Similarly, medium off-target effects can be quantified as those greater than 30% activity and low off-target effects would include all activities. Depending on the user preferences, a parameter ($\mu$) is chosen that defines the shape of the penalty distribution: a probability distribution that differentially weighs off-target inhibitor activity ranges. This penalty distribution is limited in this work to one of three shapes: tight ($\mu = 200$), medium ($\mu = 700$), or broad ($\mu = 1200$), which emphasize high, high and medium, or high, medium, and low off-target activities, respectively (*Figure 2—figure supplement 1c*).

Third, the cumulative activity of a single inhibitor or a mixture of inhibitors is calculated for each off-target kinase at the concentration of the inhibitor(s) required for 90% on-target activity. This is represented in the off-target distribution for a given inhibitor/inhibitor mixture as the kinome-wide probability to inhibit any kinase as a function of inhibitor activity. Numerous metrics describe the selectivity of single compounds (*Karaman et al., 2008*; *Klaeger et al., 2017*; *Graczyk, 2007*; *Cheng et al., 2010*; *Uitdehaag and Zaman, 2011*; *Uitdehaag et al., 2012*; *Bosc et al., 2017*; *Wang et al., 2022*); we implement an information-theoretic metric (JSD score) that best describes selectivity (see Materials and methods). The JSD score is the Jensen–Shannon distance between the off-target distribution and the penalty distribution. This score ranges from 0 to 1 and describes the overlap between the two distributions. A high score (closer to 1) represents high selectivity: only a few off-targets are in penalized activity ranges and susceptible to potent inhibition. The JSD score has similar behavior to other selectivity metrics (*Figure 2—figure supplement 2*), good reproducibility, and performs well across different underlying structures for the penalty distribution (*Figure 2—figure supplement 1a*; *Figure 2—figure supplement 1b*).

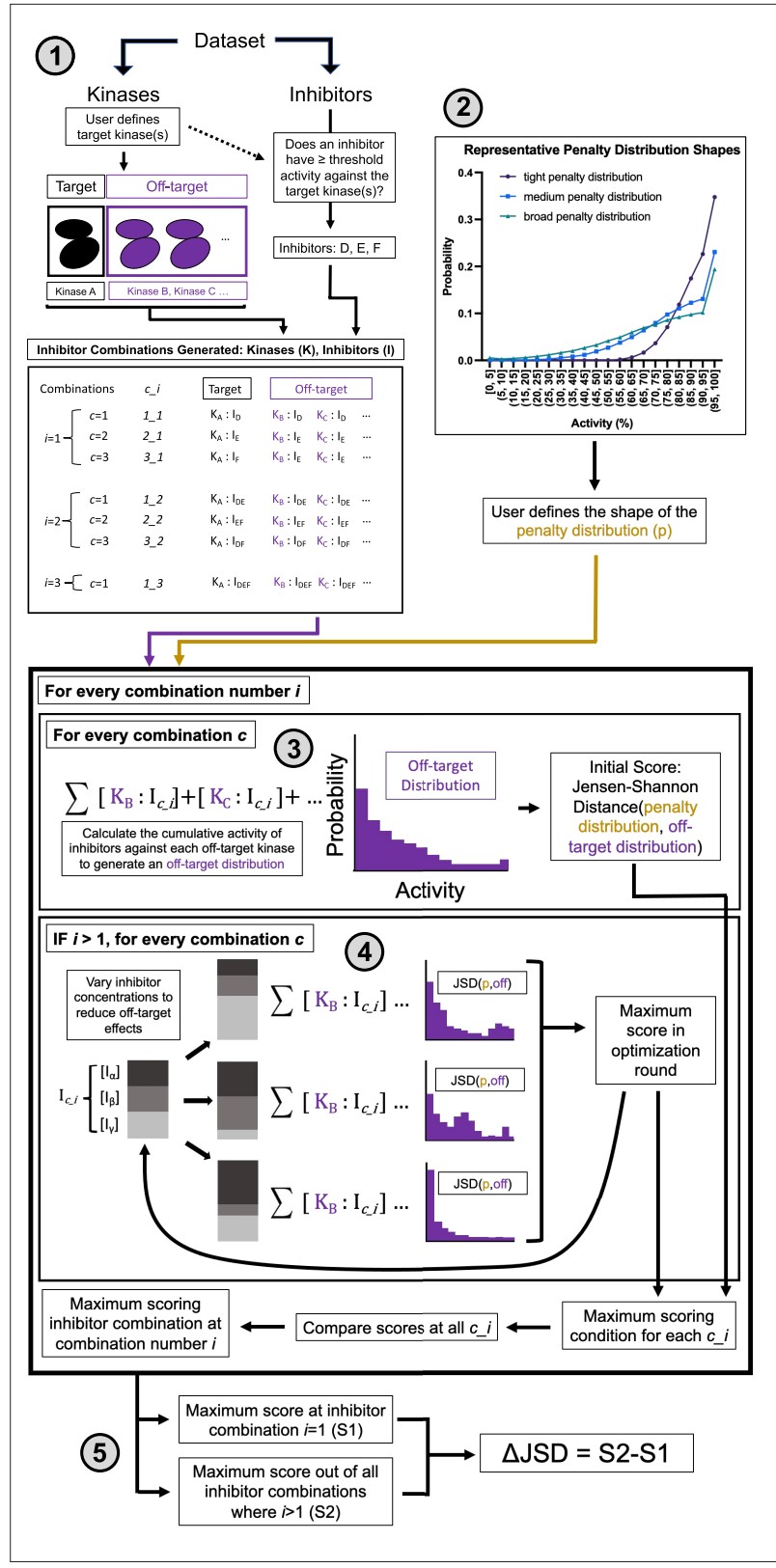

**Figure 2.** Multicompound–multitarget scoring (MMS) predicts optimally selective inhibitor combinations. (1) Combinations of sufficiently potent inhibitors are identified and enumerated where $i$ is the number of inhibitors in a combination and $c$ is a unique combination of inhibitors for each $i$. (2) A user-selected distribution shape (e.g. Poisson pmf) and associated user-selected parameters generate a penalty distribution, which defines what range

*Figure 2 continued on next page*

*Figure 2 continued*

of off-target activity will be penalized and by how much. (3) The cumulative activity of combinations of inhibitors is calculated, and (4) the relative concentration of inhibitors is adjusted to maximize the Jensen–Shannon distance (JSD) score. (5) The highest scoring combination of inhibitors is compared to the most selective single inhibitor. If the combination has a higher score (positive ΔJSD), then it is predicted to be maximally selective.

The online version of this article includes the following figure supplement(s) for figure 2:

**Figure supplement 1.** Variation of primary method parameters.

**Figure supplement 2.** Comparison of Jensen–Shannon distance (JSD) scores with other selectivity scoring methods.

**Figure supplement 3.** Imatinib selectivity for ABL1-nonphosphorylated versus ABL1-phosphorylated quantified with Jensen–Shannon distance (JSD) scores.

Fourth, the concentrations of inhibitors in every combination are optimized to maximize the JSD score. Concentrations of pairs of compounds are varied and the entire combination is rescored, and the highest scoring concentration variations are advanced through additional optimization rounds until the score no longer improves. An upper and lower limit can be enforced on the concentration range that the method will sample to enforce solubility limits, compound availability, toxicities due to reactive chemical groups, or other relevant experimental factors. Finally (fifth), the highest scoring single inhibitor and combination of inhibitors are compared (ΔJSD score). If the score of the combination is better, defined by both a statistical increase across technical replicate calculations and a minimum improvement in the magnitude of the ΔJSD score, then that combination of inhibitors is predicted to reduce off-target effects relative to the most selective single inhibitor.

To illustrate the scale of the JSD and ΔJSD scores we calculate the selectivity of imatinib for ABL1-nonphosphorylated (JSD score = 0.980) and ABL1-phosphorylated (JSD score = 0.959) using $K_d$ values from the Davis et al. dataset and the medium penalty distribution (*Figure 2—figure supplement 3*). The ΔJSD score of 0.021 arises from the ~19-fold change in imatinib affinity (ABL1-nonphosphorylated $K_d$ = 1.1 nM, ABL1-phosphorylated $K_d$ = 21 nM *Davis et al., 2011*).

The MMS approach and JSD scoring are designed to promote flexibility in user studies. The penalty distribution is used to score off-target effects in a user-defined activity range. Settings can be adjusted to meet desired thresholds for on-target potency, with different thresholds for target kinases depending upon the biological context or experimental questions being studied. Mixture selectivity can be optimized for global selectivity against all off-target kinases, certain subsets, or with different weights across off-target sets.

## Inhibitor combinations outscore single inhibitors with increasing simulated inhibitor set sizes

First, we set out to understand how the repertoire of inhibitors affects our ability to predict useful inhibitor combinations. We simulate sets of inhibitors using PKIS2-645, a large dataset of relatively selective compounds (*Drewry et al., 2017*; *Figure 3a*). The average activity distribution of PKIS2-645 compounds against all kinases reflects the average selectivity of PKIS2-645 compounds. Inhibitors simulated from this PKIS2-645 distribution have similar selectivity to true PKIS2-645 inhibitors (*Figure 3—figure supplement 1*). We generate two other activity distributions, one representing hypothetical inhibitors with slightly reduced selectivity and a third that has non-zero activity and is the least selective of the three (*Figure 3a*). Importantly, the probability of reaching threshold (90%) on-target activity is comparable between these three profiles (*Figure 3a*). We also construct a profile for binary inhibitors which either do or do not have potent (90%) activity against a target (*Figure 3a*). Sets of simulated inhibitors against 100 kinase targets are bootstrapped from these parent selectivity distributions; these simulated sets contain selectivity diversity due to random sampling and the overall profiles match expected trends of the parent profiles (*Figure 3—figure supplement 1*).

The JSD score of the best single inhibitor increases with increasing set size; as more compounds are generated there is a higher likelihood of simulating a highly selective inhibitor (*Figure 3b*). Consequently, the improvement in ΔJSD score for inhibitor combinations either remains constant or decreases with increasing inhibitor set size (*Figure 3c*). This effect is not specific to a particular penalty distribution shape (*Figure 3e*). Additionally, the minimum threshold imposed on ΔJSD scores

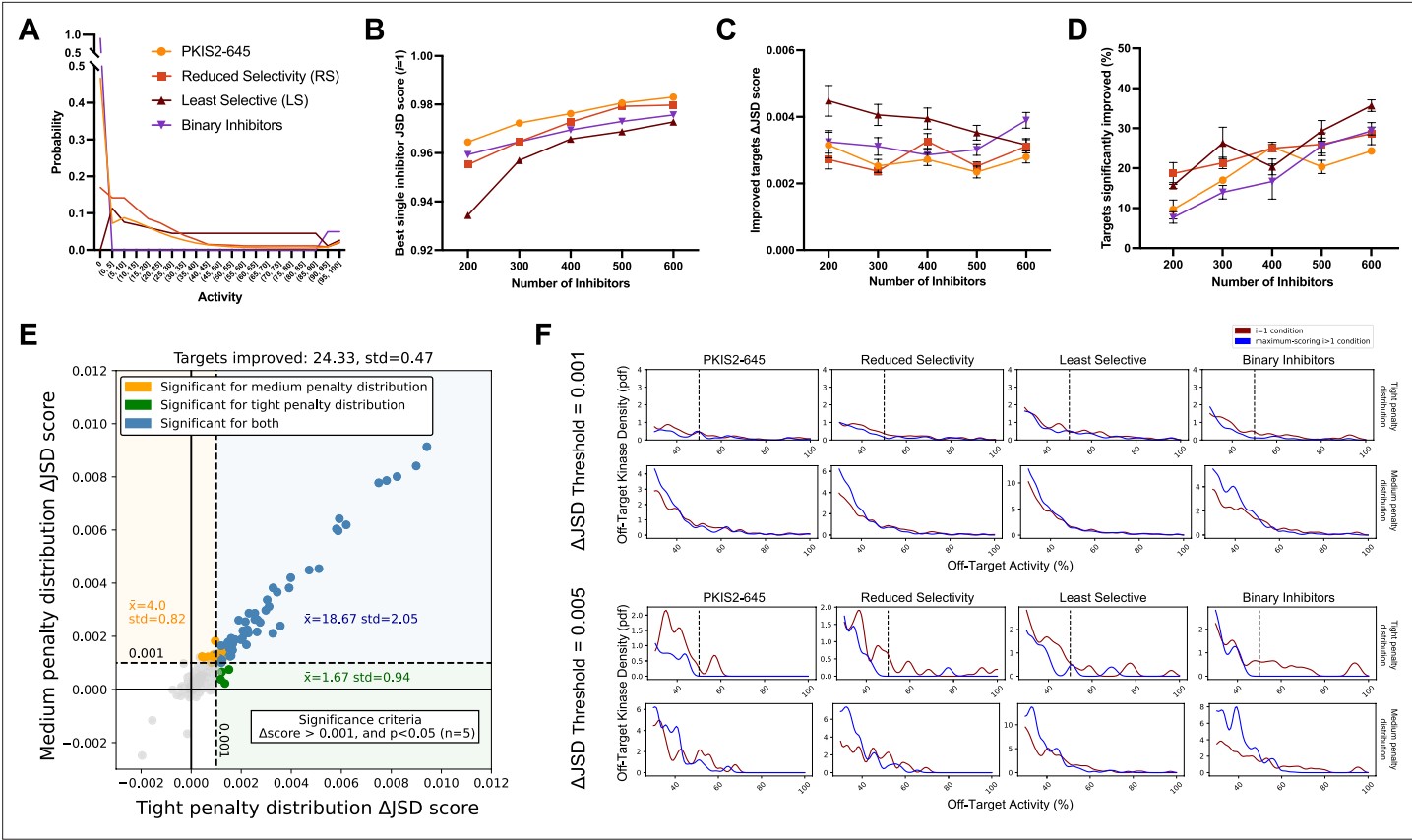

**Figure 3.** Simulated inhibitor combinations reduce off-target effects for single targets. (**A**) Simulated activity profiles for four inhibitor types. The average selectivity profile of PKIS2-645 compounds was smoothed to generate the hypothetical reduced selectivity (RS) and least selective (LS) profiles, and the binary inhibitor profile reflects hypothetical compounds with either potent (>90%) or zero activity. (**B**) As more inhibitors are simulated, increasingly selective single inhibitors are obtained. (**C**) The improvement in Jensen–Shannon distance (JSD) score either remains constant or decreases as the window for improvement narrows (max JSD = 1). (**D**) As more inhibitors are simulated, mimicking larger datasets, more kinase targets are predicted to be more selectively inhibited by a combination over the most selective single inhibitor for each target. (**E**) The improvement in off-target profiles is not specific to the shape of the penalty distribution used to calculate the JSD score. Here, three different simulated sets of 600 inhibitors generated from the PKIS2-645 parent profile are considered for 100 kinase targets. Points represent single kinase targets from three separate simulations, and the average and standard deviation of the number of kinase targets that could be significantly improved across these simulations are indicated in the figure text. (**F**) The absolute threshold used in JSD scoring corresponds to the magnitude of the improvement in selectivity; a higher threshold (0.005) selects combinations that are more selective relative to the best single inhibitor compared to combinations at a lower JSD threshold (0.001). In all cases, results from simulated analyses using three sets of 600 inhibitors are presented. The blue and red lines represent the average off-target profiles for all targets with ΔJSD scores greater than the indicated thresholds. Larger or left-shifted spaces between the red (average of the best single compounds) and blue (average of the best combinations) lines indicates a larger improvement in selectivity. The dotted vertical line at 50% activity in the plots scored using the tight penalty distribution indicate the approximate end of that penalty distribution.

The online version of this article includes the following figure supplement(s) for figure 3:

**Figure supplement 1.** Simulated inhibitors sets have selectivity diversity and overall profiles that match parent profiles.

corresponds to the magnitude of improvement in off-target profiles (*Figure 3f*). Importantly, as the set of inhibitors becomes larger and the combinatorial possibilities grow, the percentage of kinase targets that can be more selectively inhibited with a combination of inhibitors increases for all four of these selectivity profiles (*Figure 3d*). This suggests that, as more compounds are studied and aggregated into larger datasets, the utility of MMS to predict selective inhibitor combinations that outperform single inhibitors will improve.

## Inhibitor combinations are predicted to improve selectivity over the most selective single inhibitors for single kinase targets using chemogenomic datasets

We consider the optimal combination of inhibitors for all unique single targets in PKIS2-645 (*Drewry et al., 2017*) (max $i$ = 3, where $i$ is the number of different kinase inhibitors), *Klaeger et al., 2017* with 243 clinically evaluated inhibitors (max $i$ = 6), *Karaman et al., 2008* with only 38 inhibitors (max $i$ = 5), or *Davis et al., 2011* with 72 inhibitors (max $i$ = 5) (*Figure 4*). Using the PKIS2-645 inhibitors, selective inhibition of 24 unique targets (6%), can be significantly improved given both statistical and absolute ΔJSD cutoffs (*Figure 4*) with just two or three inhibitors instead of the most selective single inhibitor.

Limiting off-target effects for a subset of kinases may be an experimental goal. We consider whether it is possible to improve on the selectivity of PKIS2-645 compounds for the Eph receptor tyrosine kinase family which are promising targets in immunotherapy but contain high intrafamily structural homology (*Darling and Lamb, 2019*). Some PKIS2-645 compounds have relatively low PKIS2 activity (<30%) against EPH kinases, but the flexibility of the MMS method allows such off-target effects to be adequately scored with the broad penalty distribution. A combination of inhibitors improves the selectivity for the unique inhibition of EPHA2, EPHA3, EPHA4, and EPHA5 (*Figure 4—figure supplement 1*). This case illustrates how selectivity can be optimized for selected kinase subsets and the flexibility of user-defined penalty distributions.

MMS analyses of the Davis et al. and Karaman et al. datasets also indicated improvements in selectivity for 17 targets and 6 targets, respectively. (*Figure 4*). The Davis et al. dataset contains selectivity screening for kinase mutants in addition to wild-type constructs, including those of ABL, EGFR, FLT3, and KIT. Interestingly, multiple FLT3 alterations were found to be more selectively inhibited using a combination of inhibitors than a single inhibitor, and these inhibitors were different from those which were most selective for FLT3 (*Table 1*). Although this is a small sample size, it raises the intriguing possibility that this method may be useful in cases where selective single-compound targeting of clinically relevant mutations has not yet been actualized.

We investigate whether or not it is possible to reduce the off-target effects of clinically evaluated inhibitors, using a dataset published by Klaeger et al. This comprehensive dataset includes $K_d^{app}$ measurements for both kinase and non-kinase proteins but the data matrix is notably sparse in some areas. While PKIS2-645 assigns PKIS2 activity for 53% of compound–kinase pairs and Davis et al. reports a non-zero $K_d$ for 30% of compound–human kinase pairs, Klaeger et al. assigns a $K_d^{app}$ to just 6% of all compound–protein pairs which limits the effectiveness of off-target calculations. Nevertheless, single-target analysis suggested improvements in selectivity for kinases including EPHA5, CDK17, and PDK1 (*Figure 4*).

Single-target analysis from these datasets suggests that the single most selective inhibitor can be outperformed using a combination of just two or three inhibitors for some kinase targets. Not surprisingly, since the datasets in Davis et al., Karaman et al., and Klaeger et al. were obtained using different inhibitor sets, kinases, and methodologies, the predictions for inhibitor combinations and their impact vary between datasets.

## MMS prediction of cumulative compound activity is validated in cellulo

Next, we aimed to determine whether the MMS scoring framework applied to the in vitro PKIS2-645 dataset can predict inhibitor combinations with in cellulo activity. Compounds TPKI-108, UNC10225285A, and UNC10225404A have validated on-target action against MAPK14 (p38-alpha), with experimentally determined $K_d$s of 150, 140, and 250 nM, respectively (*Drewry et al., 2017*). These values matched the MMS 90% activity threshold of 111 nM well. MMS predicted that TPKI-108 was the most selective single inhibitor of MAPK14 ($i$ = 1), but that a combination of all three could modestly improve selectivity further.

We studied the activity of these compounds against MAPK14 and major off-targets using the in-cell NanoBRET target engagement assay (*Figure 5*; *Figure 5—figure supplement 1*). We found that the EC$_{50}$s of these compounds diverged from previously observed $K_d$s. TPKI-108 was slightly more potent (EC$_{50}$ = 82 nM), while UNC10225404A (EC$_{50}$ = 4.6 μM) and UNC10225285A (EC$_{50}$ = 1.8 μM) were approximately an order of magnitude less potent in this assay (*Figure 5a*). Single-compound activity at off-targets also differed from singlicate screen PKIS2 activity values. For example, UNC10225404A had more than 90% PKIS2 activity against MAPK14 and only 18% PKIS2 activity against MAPK11

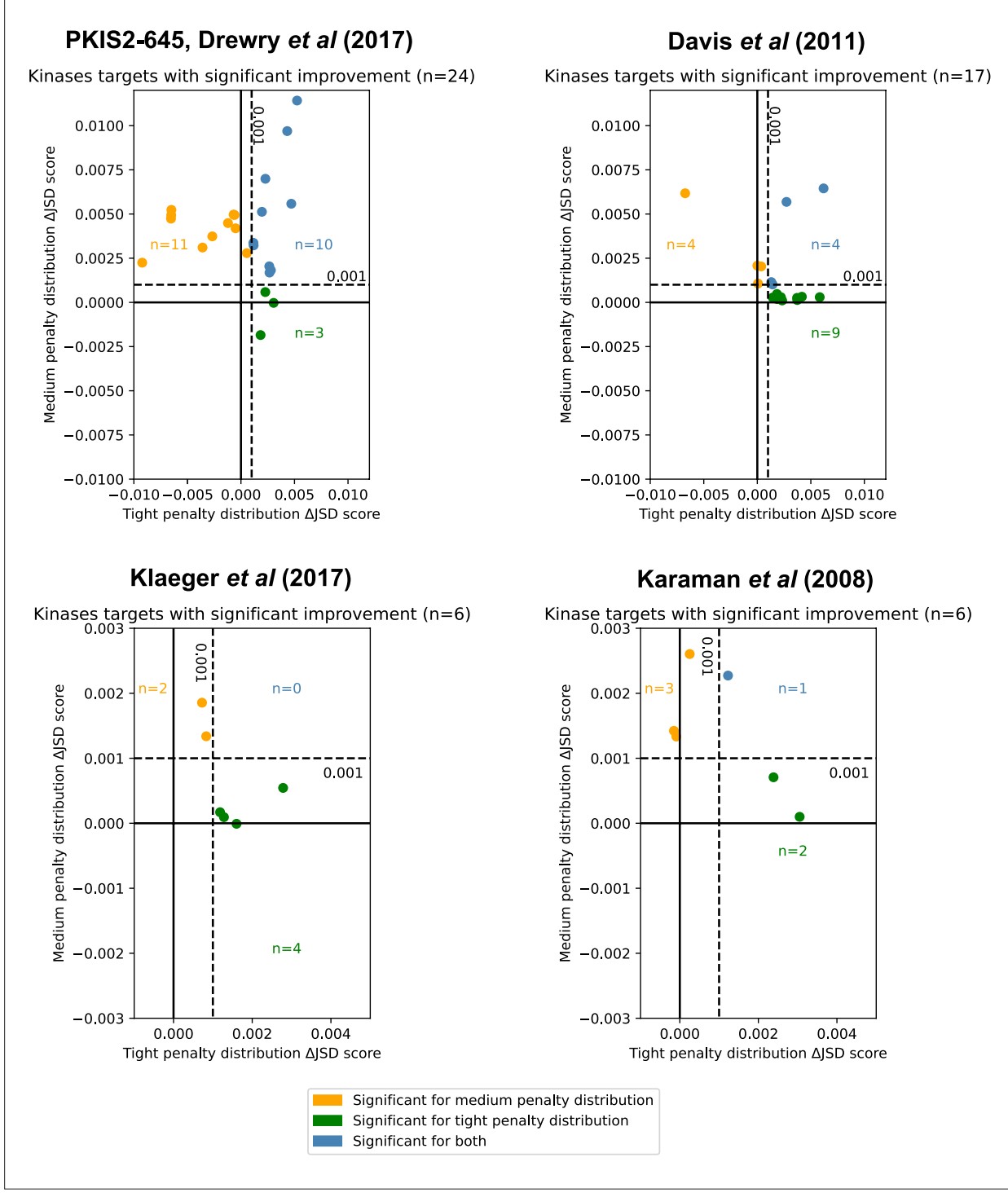

**Figure 4.** Inhibitor combinations identified using chemogenomic data reduce off-target effects for single kinase targets. Combinations of two ($i$ = 2) or three ($i$ = 3) inhibitors reduce off-target effects for some kinase targets across three chemogenomic datasets. Each point represents a single-target kinase that can be more selectively inhibited with a combination of inhibitors than with a single inhibitor. Scatterplot data are represented as the ΔJSD score using a tight penalty distribution (*x*-axis) versus the ΔJSD score using a medium penalty distribution (*y*-axis). Five technical replicates are performed for all analyses; significance was determined by an absolute improvement of greater than 0.001 in the average score across replicates, as well as statistical significance between the highest scoring $i$ > 1 and $i$ = 1 scores (*t*-test, two sided, $n$ = 5, p < 0.05). The color scheme for denoting significance is the same as in **Figure 3e**.

The online version of this article includes the following source data and figure supplement(s) for figure 4:

*Figure 4 continued on next page*

*Figure 4 continued*

**Source data 1.** Kinases with significant improvements in off-target effects using combinations of inhibitors by chemogenomic source dataset.

**Figure supplement 1.** Minor reduction in off-target effects for EPH family kinases.

(p38-beta). We found that the potency of UNC10225404A in the NanoBRET assay was quite similar against both proteins (MAPK14: $EC_{50}$ = 4.6 µM, MAPK11: $EC_{50}$ = 2.4 µM) (*Figure 5a*). About half (5/11) of the kinase off-targets with more than 50% PKIS2 activity at 1 µM with these compounds in the PKIS2-645 screen were only poorly inhibited in the NanoBRET assay ($EC_{50}$ > 50 µM for all replicates). We also conducted NanoBRET target engagement assays using cell lysates in order to validate compound–target interactions past our conservative 50 µM detection cutoff for the in cellulo assay (*Figure 5b*). These data closely matched our in cellulo results and provided additional validation of lower potency off-target $EC_{50}$s. These data suggest that different experimental techniques or the biochemical state of kinase proteins in different experimental formats may produce significant deviations in observed compound–kinase activity measurements, both for singlicate PKIS2 activity data and even multiple-point $K_d$ curves.

Next, we combined the three compounds in equimolar ratios and the cumulative activity of the mixture matched our predictions which were based on the single-compound NanoBRET $EC_{50}$ values (*Figure 5c–e*). For example, given the average three in cellulo $EC_{50}$ values of the single inhibitors against MAPK14, the cumulative activity of the equimolar combination is calculated to be $EC_{50}$ = 231 nM, very close to the average observed $EC_{50}$ = 246 nM by NanoBRET. These calculations were performed using the protocol in the MMS method. This excellent matching was observed for all kinases for which the inhibitor combination potency could be calculated. These observations strongly support our hypothesis that the cumulative activity of a combination of compounds can be predicted given the activity of each single compound against both targets and off-targets of interest.

$EC_{50}$ values from the in cellulo or lysed mode experiments were used to calculate inhibitor off-target effects if used at the minimum concentrations necessary to reach either 90%, 70%, or 50% on-target activity against MAPK14 (*Figure 5f*; *Figure 5—figure supplement 1b*). There is a modest increase in the selectivity of MAPK14 inhibition when using the combination of inhibitors over the most selective single compound, TPKI-108, which is already highly selective. Although minor off-target effects are introduced as a result of adding more inhibitors, the combination slightly improves selectivity over the primary off-target MAPK11 compared to TPKI-108.

These proof-of-concept experiments support our hypothesis that combinations of inhibitors can improve the selectivity of target inhibition by reducing high off-target effects. Even if such combinations

**Table 1.** FLT3 mutants are predicted to be more selectively inhibited with different inhibitor combinations than FLT3. Multicompound–multitarget scoring (MMS) analysis was performed with the tight penalty distribution and the *Davis et al., 2011* dataset. Concentrations reflect the amount of each respective compound necessary to reach 90% activity against the target. More than one listed combination suggests that there are multiple similar options to improve selectivity relative to the most selective single inhibitor.

| Target | Best single compound | Concentration (90% activity) | Highest scoring combination(s) | Concentrations (90% activity) |
|---|---|---|---|---|
| FLT3 | CHIR-258/TKI-258 | 5.76 nM | AC220, CHIR-258/TKI-258, R406 | 3.90 nM, 1.92 nM, 2.13 nM |
| FLT3(D835H) | BIBF-1120 (derivative) | 6.39 nM | BIBF-1120 (derivative), R406 | 3.19 nM, 2.93 nM |
| FLT3(D835Y) | BIBF-1120 (derivative) | 3.78 nM | ABT-869, BIBF-1120 (derivative), JNJ-28312141 | 9.83 pM, 3.08 nM, 2.75 nM |
| | | | BIBF-1120 (derivative), JNJ-28312141, Sunitinib | 3.08 nM, 2.75 nM, 2.06 pM |
| FLT3(ITD) | R406 | 4.86 nM | BIBF-1120 (derivative), JNJ-28312141, R406 | 2.16 nM, 0.81 nM, 3.32 nM |
| | | | BIBF-1120 (derivative), LY-317615, R406 | 2.21 nM, 1.78 nM, 3.41 nM |
| | | | BIBF-1120 (derivative), PKC-412, R406 | 2.16 nM, 2.12 nM, 3.33 nM |
| FLT3(K663Q) | PKC-412 | 18.0 nM | MLN-518, PKC-412, R406 | 2.05 nM, 10.9 nM, 1.85 nM |
| | | | AC220, PKC-412, R406 | 3.00 nM, 10.9 nM, 1.85 nM |
| FLT3(N841I) | MLN-518 | 261 nM | ABT-869, MLN-518, PKC-412 | 7.09 nM, 158 nM, 8.18 nM |

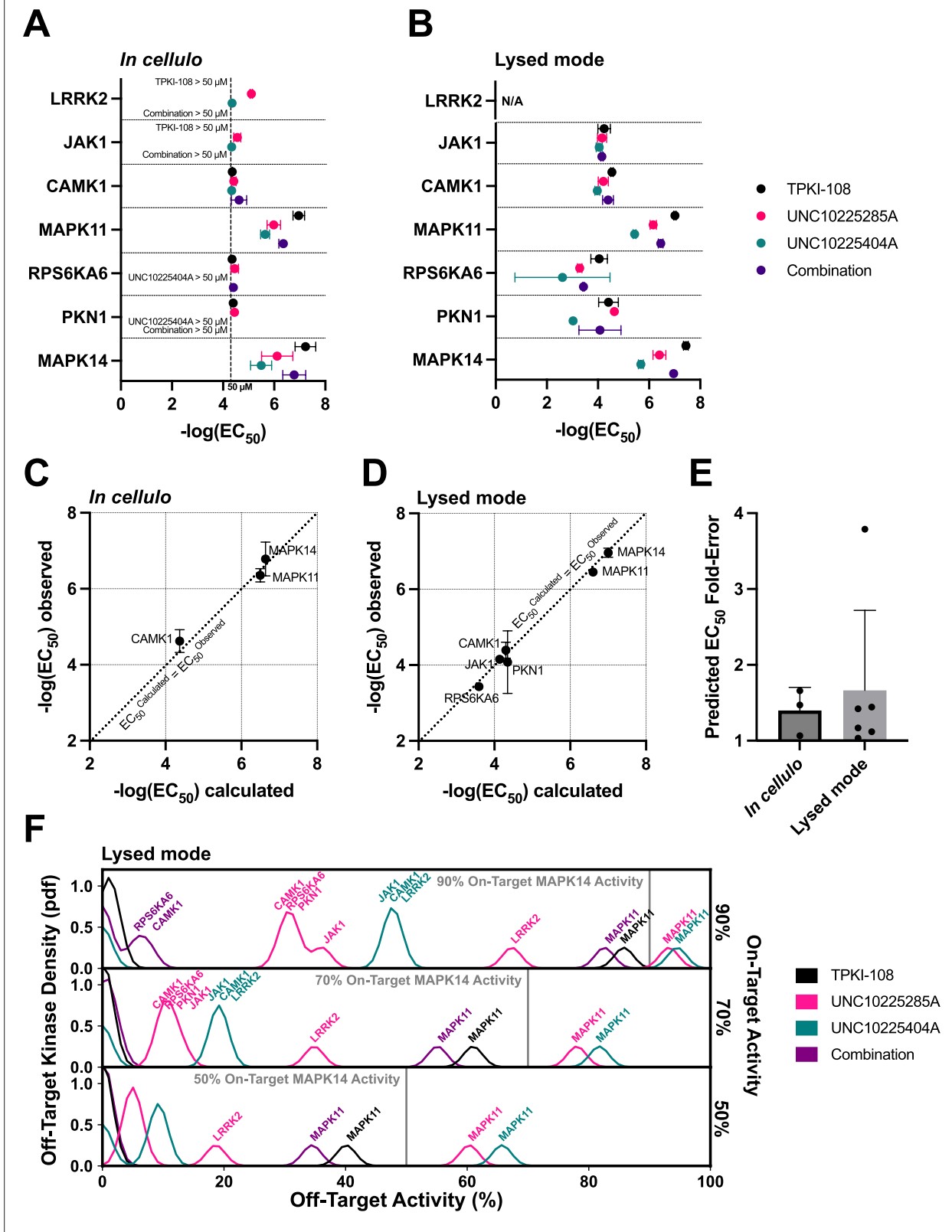

**Figure 5.** Validation of compound combination predicted activity and selectivity. The cumulative activity of each compound and equimolar combinations of compounds were studied with the NanoBRET assay in cellulo (**A**) and in cell lysates (**B**). EC$_{50}$ values were obtained for MAPK14 and those off-targets with high PKIS2 activity for any of the three single inhibitors. Error bars indicate the standard deviation. (**C, D**) The predicted cumulative activity of the three compounds, in their equimolar mixture, was calculated from the average EC$_{50}$ values of the individual compounds using the same

*Figure 5 continued on next page*

*Figure 5 continued*

protocol in the multicompound–multitarget scoring (MMS) method. (**E**) Predictions matched observed EC$_{50}$ values. Data plotted as mean and includes individual points, error bars indicate the standard deviation. (**F**) Calculation of the off-target profiles of the inhibitor combination and the individual inhibitors suggests that the combination modestly improves selectivity for MAPK14 over the most selective single inhibitor, TPKI-108. The off-target activity of the inhibitors is calculated based upon their EC$_{50}$s (lysed mode) for each off-target kinase, at the concentration needed to reach 90%, 70%, or 50% on-target activity against MAPK14. The variance of the Gaussians used to generate probability density functions (pdfs) is the same as was used in the MMS scoring method (2.5) and the activity scale is shown between 0% and 100%.

The online version of this article includes the following source data and figure supplement(s) for figure 5:

**Source data 1.** NanoBRET in cellulo and lysed mode compound and compound combination activity.

**Figure supplement 1.** Compound combination activity and selectivity in cellulo and in lysed mode NanoBRET.

add more unique off-targets, by dosing the compounds at the appropriate concentrations we expect that these minor off-target effects will be negligible. We observed that EC$_{50}$ values in the NanoBRET format varied from $K_d$ values and PKIS2 activity values determined in a separate experimental system. Consequently, predictions of cumulative compound activity would be most accurate if generated from single-inhibitor data based on the same experimental system. Importantly, the predicted cumulative activity of a combination of inhibitors very closely matched the observed activity in the NanoBRET assay. As predicted, the combination of three inhibitors modestly improved selectivity even over the already highly selective inhibitor TPKI-108. We expect that, for targets that lack highly selective single inhibitors, or in cases of multiple on-targets, the magnitude of improvement in on-target selectivity will be greater.

## MMS enables rational multitargeting by nominating inhibitor combinations

To our knowledge, no method has yet been developed to determine the selectivity of a set of inhibitors, given the cumulative effects of those inhibitors at multiple targets and off-targets. Such a method may be of particular use to those who wish to target compensatory pathways or multiple components of the same pathway. We performed a small screen of kinase pairs using the Klaeger et al. dataset which contains clinically evaluated inhibitors to evaluate MMS for rational multitargeting. We identified kinase pairs for which sets of three or four inhibitors outscored the best single inhibitor given an on-target threshold of 90% activity for both target kinases. Interestingly, the predicted improvement in selectivity (as reflected by greater ΔJSD scores) was much greater than those observed for single targets in the same dataset. Targets approached or exceeded ΔJSD >0.01, a full order of magnitude greater than the 0.001 cutoff, suggesting that the magnitude of improvement in selectivity for multiple targets might be substantial (*Figure 6a*). We analyzed the Davis et al. dataset and again observed greater improvements in predicted selectivity when targeting just two kinases rather than one (*Figure 6a*). These results encouraged further validation of MMS multitarget predictions.

In order to validate out multitargeting approach we considered kinase target sets from the Davis et al. dataset. This dataset contains $K_d$ values generated with a commercially available competitive displacement assay allowing predictions to be validated in the same assay format.

We selected a target set to study whether we could reduce high off-target effects in a translationally relevant system: ABL1 in the context of an imatinib-resistance mutation (F317L) (*Lyczek et al., 2021*; *Jabbour et al., 2008*), FYN, and LYN (*Figure 6b*; *Figure 6b, c*). FYN is a major downstream target of ABL1 in human leukemias (*Ban et al., 2008*; *Singh et al., 2012*) and LYN signaling promotes resistance to ABL1 inhibitors (*Donato et al., 2003*; *Ingley, 2012*; *Ptasznik et al., 2004*). We also selected the target pair CDC2L5 (CDK13) and TRKC (NTRK3) to test a large predicted improvement in global selectivity across a wide range of off-target activities (*Figure 6c*).

MMS analysis of the candidate trio ABL1(F317L), FYN, and LYN indicated that the most selective single inhibitor that was sufficiently potent against all three kinases was dasatinib. Masitinib was observed to be relatively selective although it lacked sufficient potency against all targets. MMS predicted that a mixture of masitinib, dasatinib, and PD-173955 (Mixture A) could be engineered to reduce high off-target effects versus dasatinib alone (ΔJSD tight penalty distribution = 0.023) while potently inhibiting all three kinases (*Figure 6c*). We tested our compound stocks in the same commercial assay format as Davis et al. We found that the observed $K_d$ values differed by less than fourfold

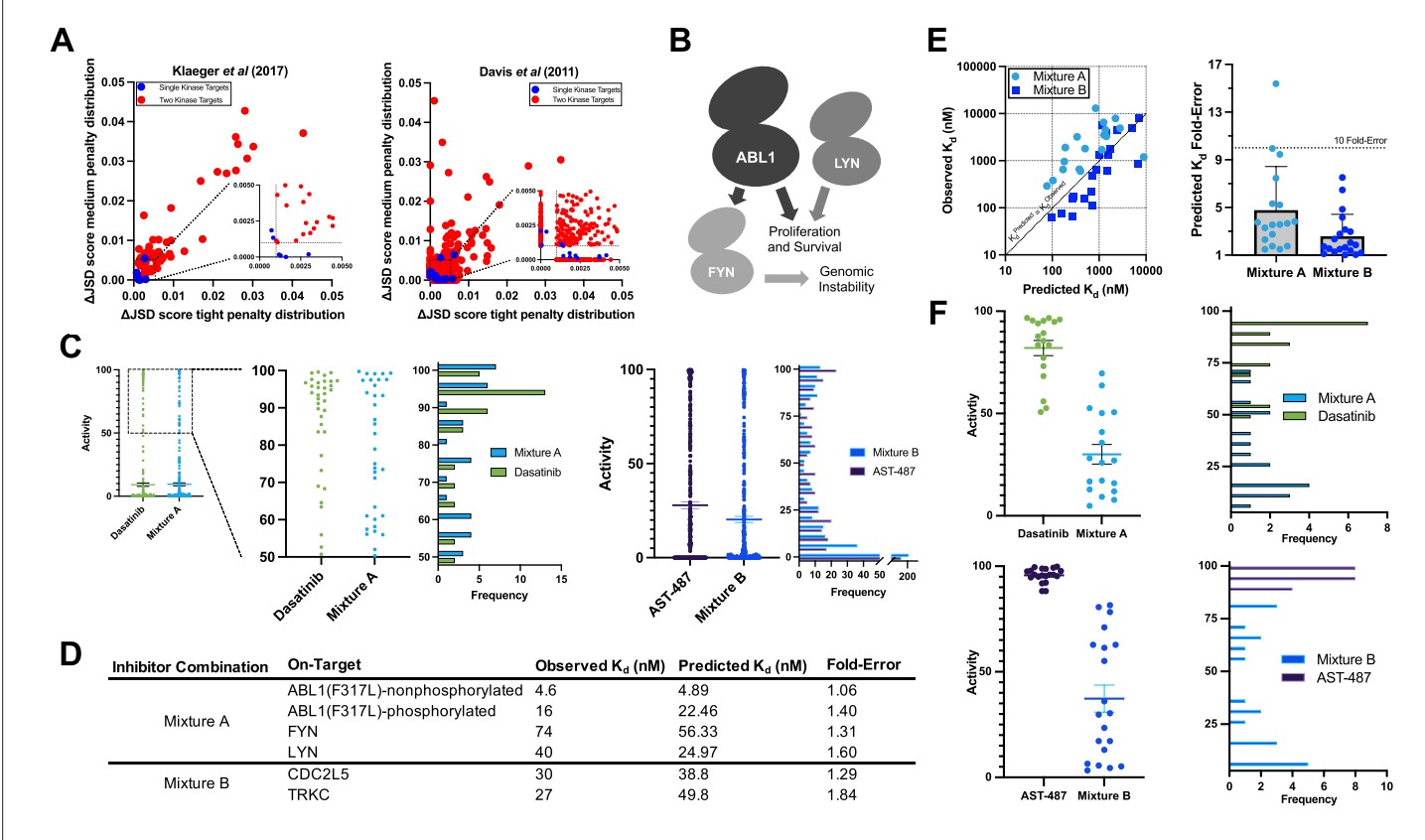

**Figure 6.** Validation of multicompound–multitarget scoring (MMS) multitarget predictions. (**A**) MMS screens suggest that off-target activity reduction may be substantially greater for multiple kinase targets compared to single kinase targets with inhibitor combinations. (**B**) ABL1, FYN, and LYN represent a translationally interesting target set. (**C**) MMS predictions suggest that high off-target effects can be reduced for ABL1(F317L), FYN, and LYN relative to dasatinib, and that global off-target effects can be reduced for CDC2L5 and TRKC relative to AST-487. Both dasatinib and AST-487 are the most selective single compounds for their respective targets out of the set of compounds with a $K_d$ of at worst 111 nM for all targets. Each point represents a single off-target kinase and the histograms summarize the adjacent dot plots. The lines in the dot plots represent the mean and the standard error of the mean. (**D**) Experimental $K_d$ values closely match predicted on-target values, and both Mixtures A and B are potent against all target kinases. (**E**) The off-target effects of both Mixtures A and B are, on average, within fivefold of predicted $K_d$ values. Each dot represents a single off-target kinase; 18 off-targets were considered for Mixture A and 20 off-targets were considered for Mixture B. Data plotted as mean and includes individual points, error bars indicate the standard deviation. (**F**) Top: Activity of dasatinib and Mixture A against 18 off-target kinases at the concentration of dasatinib (7.1 nM) and Mixture A (661 nM masatinib, 0.124 nM dasatinib, 4.69 nM PD-173955) required to inhibit ABL(F317L)-nonphosphorylated, ABL1(F317L)-phosphorylated, FYN, and LYN by at least 90%. Bottom: Activity of AST-487 and Mixture B against 20 off-target kinases at the concentration of AST-487 (306 nM) and Mixture B (4.03 nM AST-487, 226 nM ABT-869, 40.0 nM EXEL-2880/GSK-1363089) required to inhibit CDC2L5 and TRKC by at least 90%. Each dot represents a single off-target kinase, and adjacent histograms summarize the dot plots. The lines in the dot plots represent the mean and the standard error of the mean.

The online version of this article includes the following source data and figure supplement(s) for figure 6:

**Source data 1.** Inhibitor combinations are potent and more selective than the most selective single compound for multiple targets.

**Figure supplement 1.** The theoretical error of $K_d$ predictions for inhibitor combinations is limited by the maximum error in single-inhibitor $K_d$ values.

from the values published 12 years ago, indicating a high reproducibility of the commercial assay. Updated $K_d$s were used to adjust masitinib, dasatinib, and PD-173955 to final molar ratios of 5340 to 1.00 to 37.8, respectively. Mixture A was tested in the same assay format as Davis et al. against ABL1(F317L)-phosphorylated, ABL1(F317L)-nonphosphorylated, FYN, LYN, and a subset of off-targets potently inhibited by dasatinib alone. Mixture A $K_d$ values were within twofold of predictions for the four on-targets, and passed the 90% activity (111 nM) threshold (***Figure 6d***).

Next, we selected a subset of off-targets for experimental $K_d$ determination that were predicted to have less activity with Mixture A compared to dasatinib alone at the 90% on-target activity threshold. We report all off-target $K_d$ values that were tested (***Figure 6—source data 1***). The predicted off-target

$K_d$s of Mixture A were, on average, less than fivefold different from observed $K_d$ values (*Figure 6e*). The selectivity of Mixture A was excellent and greatly reduced off-target activity versus dasatinib (*Figure 6f*). The least potently inhibited on-target kinase of both dasatinib and Mixture A was FYN, so on-target activity against FYN was used as the baseline for calculating off-target selectivity. To calculate the fold-selectivity advantage for each off-target the fold-selectivity of Mixture A is divided by the fold-selectivity of dasatinib (*Figure 6—source data 1*). On average, Mixture A was 28.7-fold more selective against tested off-target kinases compared to dasatinib.

MMS predictions suggested that the most selective single inhibitor for CDC2L5 and TRKC in the Davis et al. dataset with sufficient on-target activity for both kinases was AST-487, but that a combination of AST-487, ABT-869, and EXEL-2880/GSK-1363089 (Mixture B) would substantially reduce off-target effects; the average activity across all off-targets was predicted to decrease by 7.5% (ΔJSD tight penalty distribution = 0.057, ΔJSD medium penalty distribution = 0.081) (*Figure 6c*). As previously, we retested our compound stocks in the commercial assay against CDC2L5 and TRKC and again found that all $K_d$s were within fourfold of the originally reported values. These updated on-target $K_d$ values were used to optimize the relative concentrations of the compounds and the on-target effect of Mixture B was tested against CDC2L5 and TRKC. Mixture B contained AST-487, ABT-869, and EXEL-2880/GSK-1363089 at molar ratios of 1.00 to 56.0 to 9.92. Predicted on-target $K_d$s for Mixture B were similarly accurate as for Mixture A; they were within twofold of experimentally observed $K_d$s for CDC2L5 and TRKC (*Figure 6d*). $K_d$s were determined for a subset of off-targets that were predicted to have less activity with Mixture B compared to AST-487 alone at the 90% on-target activity threshold. As for Mixture A, we report all of these experimentally determined off-target $K_d$ values. Predicted off-target $K_d$s for Mixture B were very accurate; on average there was a 2.6-fold difference between predicted and observed $K_d$ values (*Figure 6e*). The observed selectivity of Mixture B was excellent (*Figure 6f*) and there was, on average, a 171-fold selectivity advantage to using Mixture B over AST-487.

These data indicate that inhibitor combinations can more selectively inhibit kinase targets than available single inhibitors and that MMS predictions of cumulative target activity are very accurate.

## Discussion

MMS nominates inhibitor combinations that optimize selectivity for kinase targets. This method implements JSD scoring, a new flexible metric to quantify selectivity that can accommodate any number of compounds or targets. Selectivity can be calculated for different ranges of off-target activity by selecting a penalty distribution for high, medium, and low activity off-targets.

Using MMS on several publicly available kinase inhibition datasets, we predict that up to 24 single kinases could be inhibited more selectively through a mixture of inhibitors than through the single most selective inhibitor (*Figure 4*). We validate the prediction experimentally for MAPK14, a medically relevant target in cells (*Figure 5f*). The number of kinases predicted to benefit from inhibitor mixtures is highly dependent on the dataset. Analyses with simulated data indicate that larger datasets, with greater combinatorial options, may improve the utility of MMS even as more selective single inhibitors become available (*Figure 3d*). Additionally, the usefulness of MMS is likely to improve as datasets include less selective inhibitors so that more inhibitors with orthogonal selectivity profiles can be combined. Our results demonstrate that MMS predicts cumulative compound activity accurately and improves selectivity even against several single kinase targets for which selective single inhibitors exist.

Our predictions with chemogenomic data suggest that much greater gains in selectivity are observed when combining inhibitors against multiple target kinases to enable rational multitargeting. The inhibition of multiple kinases is extremely clinically beneficial, for example, in the case of sorafenib, which inhibits the serine/threonine kinase RAF and the receptor tyrosine kinases VEGFR and PDGFR (*Wilhelm et al., 2008*). We consider two validation cases: a set of translational targets (ABL1(F317L), FYN, and LYN) for which we predict reduction of high-off-target inhibition, and CDC2L5 and TRKC kinases for which we predict a major reduction in global off-target effects.

We find that MMS predicts the experimentally observed $K_d$s with high accuracy for on-targets (less than twofold difference) and only slightly lower accuracy for off-targets (three- to fivefold). Importantly, we observe that the inhibitor combination targeted at ABL1(F317L), FYN, and LYN (Mixture A) reduces the mean off-target inhibition from 82% using dasatinib to 30% (*Figure 6f*). We observe

an even more pronounced effect for targeting CDC2L5 and TRKC: Mixture B reduces the mean off-target activity from 96% to 37% (*Figure 6f*). While these results demonstrate the proof of principle for rational multitargeting of kinases through inhibitor combinations, MMS currently cannot predict beneficial drug inhibitor combinations for all kinases or kinase combinations because of the limited datasets available. For example, we base our predictions largely on the in vitro Davis et al. dataset which contains only 72 kinase inhibitors because it was to our knowledge the most complete dataset of $K_d$ values which improve the accuracy of the MMS predictions. With the dramatic advances in cellular target engagement assays (e.g. kinobeads or NanoBRET) we expect that larger datasets will eventually emerge and allow MMS to find beneficial predictions for more kinases and kinase combinations.

Our in vitro validation studies of compound mixture cumulative activity were designed to reflect one accessible protocol in user MMS implementation. Individual inhibitors could be benchmarked against a small subset of target kinases, these updated $K_d$ values could be used to refine the relative concentration of the inhibitors in mixtures, and mixtures could then be deployed in a system of study. We emphasize that the subsets of off-targets validated in this work do not describe the entire selectivity landscape of the tested inhibitor combinations; these particular off-targets were selected to illustrate the improvement in selectivity enabled by inhibitor combinations. However, the good accuracy of our predictions across these off-targets leads us to conclude that predicted off-target selectivity gains for the entire ensemble of off-targets would be similarly accurate. In MMS applications where particular off-targets are critical, we suggest that users additionally benchmark individual inhibitors against those off-targets in order to ensure the highest accuracy of MMS off-target calculations and predictions.

Notably, MMS may propose combinations with chemical probes that are not the most selective or potent against a particular target. This work offers a fresh perspective on the utility of these compounds, and additionally highlights the value of high-quality and accessible compound–target datasets. MMS also provides a clear rationale for a new paradigm in compound development; a compound may not need be maximally selective for a desired target, but instead could be designed to have an off-target profile maximally orthogonal to those of other compounds.

Importantly, even in cases where using $N$ inhibitors may be the most selective method of inhibiting an equal number ($N$) of kinases, the most selective inhibitor combination does not necessarily consist of the most selective single inhibitors against each respective kinase. MMS identifies the most selective inhibitor set in these cases, which might not be appreciated with other selectivity evaluation methods that score the selectivity of individual compounds.

Here, we detail the implementation of the MMS framework within the scope of direct target engagement for biomedical research. Extension of MMS to phenotypic cellular assays or in vivo models would require additional modeling including phosphoproteomics, pharmacodynamics, and pharmacokinetics. The global and high-off-target selectivity gains enabled by MMS lead us to hypothesize that it would be advantageous to test MMS candidate combinations on cellular phenotypes. MMS may be of particular interest for rational multitargeting of therapeutic targets, such as kinases with compensatory signaling mechanisms or those at different stages of the same signaling pathway. Lastly, we note that MMS can be implemented for targets besides protein kinases, when minimizing off-target effects are desirable and multiple interventions against the same target are available.

## Materials and methods
### MMS methodology

First, either a single kinase, or a group of multiple kinases, are defined as the target set. The inhibitors with potent activity against these targets, over a user-defined threshold, are divided into combinations of $i$ inhibitors, where $i$ is the number of inhibitors in a single combination. We can enumerate $c_i$ unique sets for each set of inhibitors with $i$ unique compounds. If the target set contains multiple kinases, combinations that do not maintain 90% activity for all target kinases are eliminated. We use an inhibitor activity threshold of 90% for all experiments, which corresponds with a single-inhibitor $K_i$ of at least 111 nM if compounds are dosed in a reference frame of 1 µM. Program settings also facilitate input of inhibitor $K_i$ values, $K_d$ values, or $K_d^{app}$ values, which are treated equivalently.

Next, the user defines a penalty distribution. This is a probability distribution that differentially penalizes off-target effects, with a maximum at 100% activity and a left-skewed tail (*Figure 2—figure*

*supplement 1c*). We compare distributions based upon two common distributions shapes, either the (1) beta pdf or (2) the left-tail of the Poisson probability mass function (pmf), given by:

1. $\frac{\Gamma(\alpha+\beta)x^{\alpha-1}(1-x)^{\beta-1}}{\Gamma(\alpha)\Gamma(\beta)}$ for $0 \leq x \leq 1$ where $\Gamma()$ is the gamma function, for shape parameters $\alpha$ and $\beta$ where $\alpha > 0$ and $\beta > 0$.
2. $\frac{\mu^k e^{-\mu}}{k!}$ where $k \geq 0$, for shape parameter $\mu$ where $\mu \geq 0$

The curvature of these distribution can be controlled by changing associated shape parameters. A 'tight' distribution will apply greater penalties to off-target activity in the corresponding $x$-axis range underlying the distribution, but will not penalize as wide a range of off-target activity as a more 'broad' distribution with a long tail. We sample 100,000 points from each underlying distribution shape to generate the actual distributions used for calculating JSD scores.

We find that, for equivalent JSD scores generated by either the beta pdf or Poisson pmf, distributions, there is only a minimal difference in the performance (*Figure 2—figure supplement 1*). Reproducibility is defined as the percentage of first-rank identical inhibitor sets obtained from technical replicates. Non-perfect reproducibility suggests that, given the associated uncertainty introduced into activity calculations, there may be multiple inhibitor combinations that achieve similar selectivity. For distributions based upon the Poisson pmf there is no correlation between result reproducibility and score, versus a minor correlation for the distributions based upon the beta pdf (*Figure 2—figure supplement 1b*). For this reason, the Poisson pmf shape is selected as the basis for all following analyses. In this work, we alter the parameter $\mu$ to produce either tight ($\mu = 200$), medium ($\mu = 700$), or broad ($\mu = 1200$), penalty distributions (*Figure 2—figure supplement 1c*).

User selection of penalty distribution parameters depends upon the off-target activity range that is most critical for a particular application. For example, if <50% off-target activity is not meaningful for a user-specific context, that user may opt to use a tight penalty distribution that is insensitive below that threshold. Alternatively, if a user wishes to consider all possible off-target activity, they might select a broad penalty distribution.

These penalty distributions all penalize off-target activity closer to 100% more than lower off-target activity, although the tight penalty distribution has the greatest penalties in the high off-target activity range. We allow users to include an additional penalty for high off-target activity in the 95–100% activity range. This high off-target penalty accounts for logarithmic fold changes in $K_i$, $K_d$, or $K_d^{app}$ that are not readily apparent in the linear activity threshold scale, and does not affect method performance (*Figure 2—figure supplement 1a*).

Inhibitor combinations may only significantly improve selectivity when measured using particular penalty distributions; in these cases, it is important to consider context, interpretation, and user goals. We conduct analyses using two penalty distributions for all datasets: a tight penalty distribution and either a medium or broad penalty distribution, and recommend that similar complementary analyses should be performed in most user cases. A single distribution could be used when a particular known range of off-target effects is of interest; however, we suggest using at least two penalty distributions if such information is not known or accessible.

Once all inhibitor combinations and the penalty distribution have been generated, the cumulative activity of all inhibitors in a specific combination $c\_i$ against each off-target kinase is calculated. The cumulative activity or occupancy of a single target by multiple compounds is given by:

3. $I^T = \frac{\sum_{j=1}^{n}\frac{I_j}{K_{ij}}}{1+\sum_{j=1}^{n}\frac{I_j}{K_{ij}}}$ where $I^T$ is the total inhibition of a kinase by $n$ inhibitors, $I_j$ is the concentration of an inhibitor $j$, and $K_{ij}$ is the $K_i$ or $K_i$ equivalent of inhibitor $j$ against the target kinase. Activity is reported as a percentage of total target occupancy $I^T$. We note that this is an approximation that does not include the effect of varying the concentration of ATP or the kinase. The equivalent combination $K_i$ (or NanoBRET EC$_{50}$) is given by:

4. $\left(\left(\frac{1}{I^T}\right)-1\right)\sum_{j=1}^{n}I_j = K_i$

Having calculated cumulative activity at a single off-target from $I^T$ a Gaussian distribution whose mean is centered around this activity value is defined for subsequent analysis. The variance of this Gaussian can be defined by the user. Each individual Gaussian is sampled from 100 times, and these sampled off-target activity values across all Gaussians are combined into a single cumulative off-target

distribution. This cumulative off-target distribution and the penalty distribution are both normalized and binned by off-target activity. Intervals of 5% are commonly used for kinase activity profiles (*Drewry et al., 2017*). Additionally, a bin size of 5% activity is an analytical compromise; too small a bin size would make scoring overly sensitive to variations in noise sampling, and too large a range would be insensitive to functionally meaningful differences in off-target profiles.

The JSD score between the penalty distribution and the cumulative off-target distribution is calculated. The JSD score describes the overlap between these two normalized distributions. This metric falls between 0 and 1, describing either identical or entirely non-overlapping distributions, respectively. A high score closer to 1, the product of minimal overlap between the off-target distribution and the penalty distribution, represents a selective inhibitor, or combination of inhibitors. Tight penalty distributions tend to produce higher scores than medium or broad distributions since a smaller activity range is included in JSD score calculation (*Figure 2—figure supplement 1b*). In some cases, there may be no off-target activity in the range that a particular penalty distribution covers. Consequently, a JSD score of 1 may sometimes reflect that a broader penalty distribution should be used.

The concentration of inhibitors in each combination $c_i$ is then optimized to maximize the JSD score. Compound concentration optimization for a single combination of compounds is performed using a simple branch-and-bound approach. In the first step of each round, the concentration of every pair of compounds is varied by $R1$-fold, one up and one down, leading to a set size of new concentration variations equivalent to the choose-two combination. The concentrations of all compounds in these new sets are reduced by iterative step sizes of $R2$ to reach minimum on-target activity, set to 90% activity in this work. Then, the JSD score is calculated, and the top-scoring set of concentrations is used as the input for the next optimization round; all other concentration variations are culled. Optimization continues until the JSD score can no longer be increased, and the highest scoring concentrations are selected.

Cutoffs can be used to limit negligibly small concentrations of inhibitors, and while we implement no maximum concentration cutoff in most analyses to account for compound solubility issues we note that such poorly potent compounds would not normally be included in MMS analyses since they would fail to reach 90% on-target activity in a reference frame of 1 µM. Decreasing this cutoff or changing the concentration reference frame could introduce these compounds, and is something that a user should consider should they choose to do so.

Lastly, the performance of the highest scoring single inhibitor at $i = 1$ is compared to the best performing combination for all other values of $i$. The maximum-scoring single inhibitor for the target(s), $i = 1$, represents the most selective single inhibitor. If a combination at $i > 1$ has a positive ΔJSD score ($JSD_{i>1} > JSD_{i=1}$), at defined compound concentrations, then the combination would produce less off-target activity than the single most selective inhibitor while retaining potent activity against the target kinase(s). We conduct five technical replicates for each experiment in order to ensure statistical significance, and use an additional absolute ΔJSD score cutoff of 0.001 to eliminate statistically significant but functionally insignificant results. Raising the ΔJSD score cutoff to 0.005 correspondingly increases the magnitude of the observed reduction in off-target activity (*Figure 3f*). It is necessary to conduct technical replicates since noise is introduced during the course of the method when Gaussians centered on calculated off-target activity values are sampled to generate cumulative off-target distributions. Additionally, the penalty distribution is also resampled between runs.

## MMS reproducibility

Increasing the variance of the Gaussian derived from a single off-target activity calculation (using $I^T$) decreases the reproducibility of program results but better accounts for uncertainty in data measurements (*Figure 2—figure supplement 1a*). Program reproducibility is defined as the percentage of first-rank identical inhibitor sets obtained from a set of technical replicates – variability in results is introduced by increasing the variance of the noise distribution, in addition to resampling the penalty distribution between technical replicates. Consequently, program reproducibility decreases with increasing values of $i$ to approximately 60% for combinations of $i = 3$ inhibitors; there is more variability in nominated inhibitor sets as the number of allowed inhibitors in a set increases. Imperfect reproducibility is an indication of uncertainty in the results, and suggests that other equally selective single inhibitors or inhibitor combinations may be available. Reproducibility is not a measure of method validity.

## MMS limitations and assumptions

Data types used in MMS predictions require certain limiting assumptions. First, chemogenomic data are often derived from kinobead competition-based assays; inhibitors are assumed to be competitive inhibitors. Second, we make reasonable substitutions of $K_d$, $K_d^{app}$, or NanoBRET $EC_{50}$ values for $K_i$ values in cumulative activity calculations when $K_i$ values are not available. However, other data types that do not describe target engagement, such as $IC_{50}$ values from cellular proliferation assays which are related to a cellular phenotype, would not be appropriate substitutions for calculating cumulative compound activity. This is because the relationship between an observed cellular phenotype and target inhibition would also involve modeling the effects of inhibiting specific signaling pathways. Third, biochemical assays, such as kinobead assays used in PKIS2-645, Karaman et al., and Davis et al., may not fully capture kinase dynamics or the role of regulatory domains in vivo. For this reason, calculations using these data should not be interpreted as a definitive prediction of in vivo kinase inhibition when treated with inhibitors. Similarly, we do not propose drug combinations for clinical applications since integration of pharmacokinetics and pharmacodynamics would be required. Fourth, kinetic effects of target engagement, such as target-specific phenotypes due to slow off-rates, are not considered. Lastly, unlike datasets that contain $K_i$, $K_d$, $K_d^{app}$, or $EC_{50}$ values, performing inhibitor concentration optimization based upon singlicate screen PKIS2 activity values may have more false positives and negatives, and the measurements may not be as precise. Given an on-target activity threshold of 90%, a difference of a few PKIS2-activity-scale percentage points for inhibitors in the 90–100% range would produce fold change differences in calculated dilutions once the inhibitor concentrations are recalibrated to 90% on-target activity. Consequently, we use equimolar ratios of inhibitors in combinations and do not perform concentration optimization steps in our analyses with PKIS2 inhibitors.

## Major MMS settings

### Penalty distribution parameters

Both a Poisson pmf and a left-tailed beta pdf can be set by program users, with control over the shape parameters using the $\mu$ parameter for the Poisson pmf or $\alpha$ and $\beta$ parameters for the beta pdf. Both distributions are scaled from 0 to 100% activity with an area under the curve (AUC) of 1. We observe in our parameter scan tests (*Figure 2—figure supplement 1*) that both shapes give reasonably similar reproducibility per score, although there may be slightly better combinatorial resolution when using Poisson distributions for analyses in which a user wants to account more for lower off-target effects. Additionally, there is less of a correlation between JSD score and reproducibility for the Poisson pmf distribution. Due to this slight difference, we select the Poisson pmf distribution for this work. We sample from this distribution 100,000 times prior to JSD calculation; since the sampled values are normalized to an AUC of 1 regardless of their size this number can be increased if desired. We do not recommend making this number too small, as this may result in the penalty distribution having odd features, such as non-decreasing step magnitudes with decreasing activity values.

### Additional high off-target activity penalty

This additional penalty actualizes one major goal of the method, to reduce high off-target activity, and helps account for the scaling differences at very high $K_d$ or $K_i$ values. For example, a single compound with a $K_i$ of 52 nM against an off-target kinase has an estimated activity of 95.05%, while another compound with a $K_i$ of 5.2 pM has an estimated activity of 99.99%, and while using a bin size of 5 both would be penalized by the same amount if no variance was introduced to the calculations. This additional penalty is normalized, along with the underlying penalty penalty distribution, to an AUC of 1. In multitarget analyses it may be necessary to increase this penalty above 0.1.

### Distribution bin size

We opt to use a bin size of 5% activity to build our penalty and off-target distributions as a reasonable balance between competing technical limitations and limitations of results interpretation. Bins of 5% have been used previously for kinase activity profiles (*Drewry et al., 2017*). We note that using a larger bin size would mask changes that a user might find meaningful. Using a smaller bin size would be more responsive to fluctuations in the off-target distribution following the addition of sampled

noise measurements, and decrease the reproducibility of results. We find that using a 5% bin size is a good balance between a normative intuition of what constitutes a meaningful change in activity, and a desire to maintain reasonable reproducibility of results following the addition of noise in the program. We also note that, since each measurement is replaced by points sampled from Gaussians, the absolute cutoffs of the bins and activity measurements close to these values (e.g. 94.8 vs 95.2 for a bin cutoff of 95) do not constitute meaningful differences in the context of program scoring, since sampling will add a similar number of counts to each bin. The bins are primarily useful for smoothing variations in sampling within normative common-sense ranges.

## Gaussian variance

The default variance for the Gaussian from which off-target activity values are sampled is 2.5, which complements the program bin size of 5% activity. Increasing the variance decreases the reproducibility of the results generated by the program, or increases the likelihood that alternate equally selective compounds or combinations of compounds will be identified. However, variability may be desirable as it reflects uncertainty in input data measurements and calculations.

## On-target inhibition threshold

We maintain an on-target inhibition threshold of 90% activity which represents potent inhibition of target kinases. This threshold controls several points in the method, including how many compounds are initially selected either as single agents or for consideration in combinations, how far compound combinations are diluted if they have greater than threshold activity, and the minimum activity that must be maintained during compound concentration optimization steps. Decreasing this threshold increases the number of possible compound combinations and will increase processing time.

## Optimization step size

We primarily use optimization steps of $R1 = 5$ and $R2 = 1.1$. We find that an $R1$ value of 2–5 is useful; larger $R1$ values decrease processing time but reduce the precision of the final concentrations that are identified by the method. Similarly, larger $R2$ values ($>1.5$) decrease processing time but also decrease accuracy. For example, identically selective compounds would yield the same JSD score if both are dosed at concentrations that reach 90% on-target activity, but one would have a lower score if it is dosed at a higher concentration and reaches 92% on-target activity. A lower $R2$ value ensures that all compounds or compound concentrations get as close as possible to the 90% on-target threshold and ensures accurate selectivity scoring.

## Dataset sources

*Karaman et al., 2008*, *Davis et al., 2011*, and Drewry et al. PKIS2-645, 2017 were generated using a competitive displacement assay, previously described in *Fabian et al., 2005*. This is the same commercial assay used to validate multitarget predictions in this work. PKIS2-645 reports single-concentration screening values, while Karaman et al. and Davis et al. both report $K_d$s derived from multiple-point curves. PKIS2-645 contains data for 645 inhibitors and 406 targets, Karaman et al. contains data for 38 inhibitors and 317 targets, and Davis et al. contains data for 72 inhibitors and 442 targets. Klaeger et al. implemented a profiling campaign for 243 inhibitors against 343 targets using cell lysates, kinobeads, and quantitative mass spectrometry. Klaeger et al. note that kinase–inhibitor interactions scored using their kinobead assay were generally also observed in assays using recombinant proteins, but that the opposite was not necessarily true. These observations are relevant to our study because of the discrepancy in off-target coverage space between datasets. Klaeger et al. provide some interpretation of these results, describing how there might be differences in the activation state of kinases between systems, or that there could be nonspecific binding of proteins at high concentrations from cell lysates in their assay.

## Dataset preparation

The Karaman et al., Davis et al., and Klaeger et al. datasets of compound $K_d$ or $K_d^{app}$ (nM) were converted to activity values at a reference concentration of 1 μM using the approximation: activity = $100\%/[(K_d/1000\ (M)) + 1]$. A $K_d$ of 100 nM corresponds to an activity approximation of 91%, just above the lower threshold of what is considered a potent compound (90%) in our analyses. The reference

frame of 1 µM does not influence JSD scoring and was only used to standardize input; activity values were converted back into the original $K_d$ or $K_d^{app}$ values by the program during calculations. The reference frame is related to the on-target threshold. For example, if a reference frame of 100 nM was selected instead, then compounds would need to be tenfold more potent to reach 90% activity and only compounds with on-target $K_d$s better than 11.1 nM (rather than 111 nM in the reference frame of 1 µM) would be considered.

## Single-target MMS analyses

Single-target analyses was performed as following. Program settings were: tight penalty distribution $\mu = 200$, medium penalty distribution $\mu = 700$, broad penalty distribution $\mu = 1200$, sample size = 100,000, optimization steps $R1 = 5$, $R2 = 1.1$, on-target threshold = 90, noise distribution variance = 2.5, high off-target penalty = 0.1. Five technical replicates were performed. Significance was assessed by both statistical significance (t-test, p < 0.05) and an absolute JSD score cutoff. A cutoff of 0.001 was used for all analyses, except when indicated (at 0.005) during simulated data analysis. Kinase mutants in all datasets, while considered as targets for selectivity optimization, were excluded from all calculations of off-target distributions. FLT3 mutations were studied with $R1 = 2$, $R2 = 1.1$.

## Multiple-target MMS analyses

A non-exhaustive screen of kinase target pairs from the Klaeger et al. dataset was performed using the following settings changes: optimization steps $R1 = 3$, $R2 = 1.1$, and high off-target penalty = 0.3. Kinases with less than a 0.95 single-target JSD score using the medium penalty distribution were included in the analysis. A screen of kinase target pairs from the Davis et al. dataset was similarly performed: optimization steps $R1 = 4$, $R2 = 1.1$, and high off-target penalty = 0.3. Selected target sets were tested using smaller step sizes to increase the precision of concentration optimization and a conservative maximum concentration limit of 1 µM was implemented to avoid solubility limits for in vitro validation experiments.

## Eph family analysis

PKIS2-645 was reduced to only the data for the EPH kinase family, such that only off-target effects for other EPH kinases would be considered for each respective on-target EPH kinase. These data were analyzed using the single-target protocol, with a tight penalty distribution ($\mu = 200$), and a broad penalty distribution ($\mu = 1200$), instead of the medium penalty distribution ($\mu = 700$), in order to capture changes in low off-target effects.

## Comparison of JSD scoring with other selectivity scoring metrics

There are multiple approaches to score selectivity (*Karaman et al., 2008*; *Klaeger et al., 2017*; *Graczyk, 2007*; *Cheng et al., 2010*; *Uitdehaag and Zaman, 2011*; *Uitdehaag et al., 2012*; *Bosc et al., 2017*; *Wang et al., 2022*). We consider two different approaches to scoring selectivity: relative selectivity terms (e.g. partition index, CATDS score, and entropy score) that compare on-target inhibition relative to off-target inhibition; and non-relative terms (e.g. Gini coefficients, Window score, and S-score) that describe inhibitor behavior toward a set of targets without the definition of on- and off-targets. Relative selectivity scores do not distinguish between inhibitors with many moderate off-target activities and those that have a few high-off-target activities balanced with many low off-target activities. Similarly, non-relative terms like Gini coefficients do not discriminate between off-target assignments that yield equivalent scores even if a larger number of off-targets have relatively high off-target activity. In other words, a compound may not be useful if it has a small set of high off-target effects, even if the average inhibition of all off-targets is quite low. A window score (*Bosc et al., 2017*) or S-score (*Karaman et al., 2008*) integrate user-defined compound affinity ranges, since only certain magnitudes of off-target effects may be meaningful in the context of an experimental setting. However, these two scoring methods cannot differentiate between low and high off-target effects within set thresholds.

Our goal was to design a selectivity metric that would score off-target activity in the context of user-defined on-target activity. JSD scoring fulfils these criteria. Additionally, by integrating user-defined penalty distributions, off-target activities of different magnitudes can be more precisely scored than with other selectivity scoring metrics. These two features make the JSD score the ideal metric for

determining the most selective combination of inhibitors for a given target or set of targets in a particular user-defined experimental setting.

The JSD score was calculated for all single-compound single-target interactions with ≥90% on-target activity across the PKIS2-645, Davis et al., and Klaeger et al. datasets, using the tight, medium, and broad penalty distributions (*Figure 2—figure supplement 2*). A simple relative selectivity factor was calculated for each of these compound–target interactions: the activity against the target divided by the total activity of the compound at all possible targets. CATDS scores, while specific to the kinobead based assay employed by Klaeger et al., describe the decrease in compound binding for a target relative to the total reduction in compound binding across all possible targets, at a particular compound concentration. The CATDS score has, in general principle, the same structure as the simple relative selectivity factor.

JSD scoring is able to parse the selectivity of compound–target interactions among the set with poor relative selectivity factors (*Figure 2—figure supplement 2*). In other words, a promiscuous compound may still be relatively specific at some targets; JSD scoring identifies these cases while the relative selectivity scoring metric does not perform as well. Compounds that have high relative selectivity factors also have very high JSD scores (*Figure 2—figure supplement 2*). Fold-differences in the selectivity of a compound at off-targets may not affect the activity of a compound at those off-targets if off-target sites are already nearly empty (*Figure 1—figure supplement 1*). In other words, there may only be a very minor advantage to using an extraordinarily selective compound instead of an already very selective compound. While relative selectivity factors would strongly suggest using one compound over another, JSD scoring correctly identifies that there may be no or only minimal improvement possible.

Gini coefficients and S-scores (50% activity) were calculated for each inhibitor included in JSD scoring (*Figure 2—figure supplement 2*). A high Gini coefficient suggests high selectivity, while a low S-score suggests high selectivity. Unlike JSD scores or the simple relative selectivity factor, these scores are not target specific; the same score is assigned to all interactions made by the same compound producing horizontal striations on the associated plots. The striations are more pronounced for analyses with the Davis et al. and Klaeger et al. datasets since on-target concentration optimization was performed resulting in greater differences in off-target profiles after compound concentrations had been calibrated. High JSD scores are possible for some compound–target pairs even if the compounds score poorly with these selectivity metrics since using nonselective compounds at the optimal concentration for their most potently inhibited targets can still produce relatively selective outcomes.

## NanoBRET target engagement assay

The assay was performed as described previously (*Vasta et al., 2018*). In brief, full-length kinase ORF (Promega) cloned in frame with a NanoLuc-vector (as indicated in table below) was transfected into HEK293T cells using FuGENE HD (Promega, E2312) and proteins were allowed to express for 20 hr. Serially diluted inhibitor and NanoBRET Kinase Tracer (as indicated in the table below) were pipetted into white 384-well plates (Greiner 781 207) using an ECHO 550 acoustic dispenser (Labcyte). The corresponding transfected cells were added and reseeded at a density of $2 \times 10^5$ cells/ml after trypsinization and resuspension in Opti-MEM without phenol red (Life Technologies). The system was allowed to equilibrate at 37°C for 2 hr and 5% $CO_2$ prior to BRET measurements. To measure BRET, NanoBRET NanoGlo Substrate + Extracellular NanoLuc Inhibitor (Promega, N2160) were added as per the manufacturer's protocol, and filtered luminescence was measured on a PHERAstar plate reader (BMG Labtech) equipped with a luminescence filter pair (450 nm BP filter (donor) and 610 nm LP filter (acceptor)). Competitive displacement data were normalized and then plotted using GraphPad Prism 9 software using a normalized 3-parameter curve fit with the following equation: $Y = 100/(1 + 10^{(X - \mathrm{LogIC50})})$. For the PRKCQ assay, the protein was stimulated by adding a final concentration of 1 µM phorbol 12-myristate 13-acetate (Sigma #P8139) in dimethyl sulfoxide.

| Target | Nluc Placement | Target Catalog No | Tracer | [Tracer], [M] | Tracer Catalog No |
|--------|----------------|-------------------|--------|---------------|-------------------|
| MAPK14 | C | NV1661 | K4 | 3.10E−08 | N2540 |
| PKN1 | C | Kind gift of Promega | K16 | 1.30E−07 | Kind gift of Promega |

*Continued on next page*

*Continued*

| Target | Nluc Placement | Target Catalog No | Tracer | [Tracer], [M] | Tracer Catalog No |
|---|---|---|---|---|---|
| RPS6KA1 | N | NV1981 | K10 | 6.30E−08 | N2840 |
| RPS6KA6 | N | NV2021 | K10 | 1.30E−07 | N2840 |
| MAPK11 | N | NV1651 | K4 | 1.30E−07 | N2540 |
| ABL1 | N | NV1011 | K4 | 1.30E−07 | N2540 |
| BRAF | C | NV2481 | K10 | 1.00E−06 | N2840 |
| CAMK1 | N | NV2531 | K9 | 6.60E−07 | N2830 |
| JAK1 | C | Kind gift of Promega | K10 | 2.50E−07 | N2840 |
| KIT | C | NV1491 | K4 | 6.30E−08 | N2540 |
| LRRK2 | C | NV3401 | K9 | 8.30E−09 | N2830 |
| PRKCQ | C | Kind gift of Promega | K10 | 5.00E−07 | N2840 |

## Cell lines

HEK293T cells from ATCC (#CRL-3216) were used in this work. Identity was confirmed via STR profiling by ATCC and the cells tested negative for mycoplasma.

## In vitro compound and mixture activity profiling

Single compound and compound mixture $K_d$s were obtained using the Eurofins DiscoverX KINO-MEscan KdELECT kinase assay product. The same compound mixture stocks were used for target and off-target $K_d$ determination to minimize sample variation. The $K_d$ values for all targets and off-targets tested with inhibitor combinations are reported in *Figure 6—source data 1*. The average of two runs are reported as a single $K_d$ value. Fold-error is defined as the maximum of either the predicted or observed $K_d$ value divided by the other; a fold-error of 1 represents a perfect prediction.

The compounds used for in vitro assays in this work were:

| | Lot # | Source | CAS # |
|---|---|---|---|
| Dasatinib | 118523 | MedChemExpress | 302962-49-8 |
| PD-173955 | 123657 | MedChemExpress | 260415-63-2 |
| Masitinib (AB-1010) | 113139 | MedChemExpress | 790299-79-5 |
| AST-487 | 7302 | MedChemExpress | 630124-46-8 |
| Foretinib (GSK-1363089) | 23056 | MedChemExpress | 849217-64-7 |
| Linifanib (ABT-869) | 18201 | MedChemExpress | 796967-16-3 |

## Fold-error simulation

Combinations of one, two, three, or four compounds are considered. Within each combination, the $K_d$ of individual compounds is randomly varied by a fold-error between one and four. This fourfold reference limit was determined empirically from limited on-target compound $K_d$ benchmarking. The cumulative $K_d$ of the combination is calculated (the observed $K_d$) and compared to what the $K_d$ of the combination would have been without the additional errors introduced (the predicted $K_d$). At each combination number one thousand cases are simulated. The distribution of fold-errors for the combination $K_d$ predictions are illustrated in *Figure 6—figure supplement 1*.

## Figure production

Figures in this work were made with Matplotlib (*Hunter, 2007*), GraphPad Prism, and KinMap (*Eid et al., 2017*). The Human Kinome dendrogram images were reproduced courtesy of Cell Signaling Technology.

## Acknowledgements

MAS acknowledges funding by NIH R35GM119437. IRO is supported by NIH T32GM136572 and NIH T32GM008444. JDC and SS acknowledge funding from MSKCC and NIH grant R01GM121505. SS is a Damon Runyon Quantitative Biology Fellow supported by the Damon Runyon Cancer Research foundation (DRQ-14-22). SK and B-TB are grateful for support from the SGC, a registered charity (no. 1097737) that receives funds from AbbVie, Bayer, Boehringer Ingelheim, the Canada Foundation for Innovation, Eshelman Institute for Innovation, Genentech, Genome Canada through Ontario Genomics Institute (OGI-196), EU/EFPIA/OICR/McGill/KTH/Diamond, Innovative Medicines Initiative 2 Joint Undertaking (EUbOPEN grant 875510), Janssen, Merck, Merck & Co, Pfizer, Takeda, and Wellcome; SK from the German translational cancer network (DKTK) and the Frankfurt Cancer Institute (FCI); and SK and B-TB from the collaborative research center 1399 'Mechanisms of drug sensitivity and resistance in small cell lung cancer'. We would like to thank Dr. David Drewry and Dr. Mohammad Anwar Hossain at the Structural Genomics Consortium and Division of Chemical Biology and Medicinal Chemistry, UNC Eshelman School of Pharmacy, University of North Carolina at Chapel Hill, Chapel Hill, for providing the compounds TPKI-108, UNC10225285A, and UNC10225404A.

## Additional information

### Competing interests

Benedict-Tilman Berger: is the CEO and a shareholder of CELLinib GmbH, Frankfurt, Germany. John D Chodera: is a current member of the Scientific Advisory Boards of OpenEye Scientific Software, Interline Therapeutics, and Redesign Science. The Chodera laboratory receives or has received funding from the National Institute of Health, the National Science Foundation, the Parker Institute for Cancer Immunotherapy, Relay Therapeutics, Entasis Therapeutics, Silicon Therapeutics, EMD Serono (Merck KGaA), AstraZeneca, Vir Biotechnology, XtalPi, Interline Therapeutics, and the Molecular Sciences Software Institute, the Starr Cancer Consortium, the Open Force Field Consortium, Cycle for Survival, a Louis V. Gerstner Young Investigator Award, and the Sloan Kettering Institute. A complete funding history for the Chodera lab can be found at http://choderalab.org/funding. The other authors declare that no competing interests exist.

### Funding

| Funder | Grant reference number | Author |
| --- | --- | --- |
| National Institutes of Health | R35GM119437 | Markus A Seeliger |
| National Institutes of Health | T32GM136572 | Ian R Outhwaite |
| National Institutes of Health | R01GM121505 | John D Chodera |
| Damon Runyon Cancer Research Foundation | DRQ-14-22 | Sukrit Singh |
| National Institutes of Health | T32GM008444 | Ian R Outhwaite |
| Structural Genomics Consortium | | Stefan Knapp |
| German Translational Cancer Network | | Stefan Knapp |
| Deutsche Forschungsgemeinschaft | 1399 | Benedict-Tilman Berger Stefan Knapp |

The funders had no role in study design, data collection, and interpretation, or the decision to submit the work for publication.

## Author contributions

Ian R Outhwaite, Data curation, Software, Formal analysis, Validation, Investigation, Visualization, Methodology, Writing - original draft, Writing – review and editing; Sukrit Singh, Software, Validation, Methodology, Writing – review and editing; Benedict-Tilman Berger, Formal analysis, Validation, Investigation, Writing – review and editing; Stefan Knapp, John D Chodera, Supervision, Funding acquisition, Writing – review and editing; Markus A Seeliger, Conceptualization, Supervision, Funding acquisition, Methodology, Project administration, Writing – review and editing

## Author ORCIDs

Ian R Outhwaite ⓘ http://orcid.org/0000-0003-2037-3261
Sukrit Singh ⓘ http://orcid.org/0000-0003-1914-4955
Stefan Knapp ⓘ http://orcid.org/0000-0001-5995-6494
John D Chodera ⓘ https://orcid.org/0000-0003-0542-119X
Markus A Seeliger ⓘ http://orcid.org/0000-0003-0990-1756

## Decision letter and Author response

Decision letter https://doi.org/10.7554/eLife.86189.sa1
Author response https://doi.org/10.7554/eLife.86189.sa2

---

# Additional files

## Supplementary files
• MDAR checklist

## Data availability

Instructions to run MMS, code, datasets, and MMS results are available on GitHub (copy archived at *Outhwaite, 2023*).

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
