## [Editor Report]

This study presents a valuable finding on a multi-compound-multitarget scoring (MMS) method that combines inhibitors to maximize target inhibition and to minimize off-target inhibition. The strategy may enable the optimization of inhibitor combinations for multiple on-targets. The evidence supporting the claims of the authors is solid. The work will be of interest to pharmacology scientists working in both academic and industrial sectors.

---

## [Decision Letter]

**Decision letter after peer review:**

Thank you for submitting your article "Death by a Thousand Cuts – Combining Kinase Inhibitors for Selective Target Inhibition and Rational Polypharmacology" for consideration by *eLife*. Your article has been reviewed by 3 peer reviewers, and the evaluation has been overseen by a Reviewing Editor and Amy Andreotti as the Senior Editor. The reviewers have opted to remain anonymous.

Essential revisions:

1) The manuscript lacks a proper validation of the proposed methodology. The authors should assess at least one combination of kinase inhibitors in the same assay format they use as the original dataset.

2) The paper is not well organized and not easily readable. The whole manuscript should be reorganized in a more concise and logical way.

3) The authors should carefully explain and elaborate in the main text why the proposed methodology presents a real advantage over the most selective inhibitor.

4) The authors should conduct a comparative analysis of their methods with other similar methods to highlight the real advantage.

*Reviewer #1 (Recommendations for the authors):*

The authors should assess at least one combination of kinase inhibitors in the same assay format they use as the original dataset. This is the only way to properly determine how well this method can work. The have been many reported differences between biochemical and cellular kinase assays and the formats should not be used interchangeably for this work.

The strength of this method relates to the application of kinase inhibitor cocktails to the study of biology. The discussion of polypharmacology should be removed. The authors cite that PK/PD would be needed to predict drug combinations in patients. Unless the authors are interested in looking at these factors, including CYP inhibition and induction to assess potential drug-drug interactions, this is too preliminary to include in the manuscript. Furthermore, there is no biological support for the polypharmacology claims – the authors would need to show that their proposed inhibitor combinations have a pharmacological benefit.

*Reviewer #2 (Recommendations for the authors):*

Please, find below some more specific comments that, I hope, might help the authors improve the quality and clarity of their manuscript.

The whole development of the method is based on the PKIS-2 set, however, it is not described in the manuscript, beyond saying that it contains 645 inhibitors for 406 kinases. Moreover, most of the data shown comes from simulations, where the authors model the "average" inhibitor in the PKIS-2, but I am sure that there is a wide heterogeneity, including kinases with many inhibitors and others with only one. Similarly, the range of on- and off-targets hit by each inhibitor must also be very heterogeneous. Without seeing a deep analysis of the PKIS-2 dataset, it is difficult to understand the validity of the simulations.

Related to the point above, the applicability of the methods seems really low, with very few kinases being able to minimally benefit from the combination of inhibitors.

The only validation of the strategy is presented in Figure 5, where the authors use nanoBRET in cells and cell lysates to quantify whether a combination of inhibitors could maintain the inhibition level on MAPK14 while reducing the off-targets binding. As shown, the most difficult cross-binding to avoid is with MAPK11 and, in this case, the superiority of the combination is very minor (if any). Indeed, the combination slightly decreases the binding with MAPK11, but it also goes down for the intended target.

I also find the text difficult to follow. In addition to the PKIS-2 data, the authors also play with other kinase-inhibitor sets, but they are not properly described and it is difficult to understand if they are being used as a support or to fine-tune the methodology. Additionally, I also find some of the figures hard to interpret (i.e. off-target kinase density vs off-target activity). The concept is pretty simple, and the authors should find a clearer way to communicate their results.

*Reviewer #3 (Recommendations for the authors):*

A few other notices.

1. The formats of "Figure " and "Figure" can be unified throughout the paper.

2. Would it be possible to create a statistical table summarizing the subfigures from Figure 3-supplemental 1 to 6, as well as Figure 4-supplementals? This would facilitate the authors' understanding of the results and enable them to grasp the meaning.

[Editors’ note: further revisions were suggested prior to acceptance, as described below.]

Thank you for resubmitting your work entitled "Death by a Thousand Cuts through Kinase Inhibitor Combinations that Maximize Selectivity and Enable Rational Multitargeting" for further consideration by *eLife*. Your revised article has been evaluated by Amy Andreotti (Senior Editor) and a Reviewing Editor.

The manuscript has been improved but there are some remaining issues that need to be addressed, as outlined below:

One of the reviewers has raised concerns on the analysis and interpretation of the extra conducted results as below. The authors should carefully discuss the results obtained, in particular for the concept of "rational multi-targeting". One referee has recommended to split it out. Please amend the manuscript accordingly (introduction, discussion and conclusions, etc).

*Reviewer #2 (Recommendations for the authors):*

I thank the authors for their answers to my comments. However, to be honest, I am a bit puzzled. The whole article was about the possibility of achieving a higher kinase specificity (and fewer off-target effects) with combinations of kinase inhibitors than with the most specific inhibitor for a given kinase. My main criticism (also highlighted by other Reviewers) is that it seems like a good idea, but the results do not support the hypothesis. Moreover, the applicability seems very limited to only a few kinases. Now, without really changing the bulk of the article, the authors have included a new figure (Figure 6) and it seems that the power of the MMS method is to identify drug combinations that are "more specific" in inhibiting combinations of kinases. However, obviously, the available inhibitors were never meant to do this function. Besides, the advantage of inhibiting two "random" kinases is very uncertain. As the paper is now, it seems like if the authors introduce a problem (i.e. low specificity of kinase inhibitors) that they cannot solve, and then change the question and find some examples where the methodology works. I am sorry not to be more positive but, in my opinion, the first version of the paper described a good idea that did not work. This second version reads like they are trying to conceal the results to sell a different success story. Again, in my opinion, it was a good idea worth testing, and it would be more honest to openly discuss the results obtained. Now this is a different paper, with a different focus, title, etc, not a resubmission. And should be treated as such.

---

## [Author Response]

Essential revisions:1) The manuscript lacks a proper validation of the proposed methodology. The authors should assess at least one combination of kinase inhibitors in the same assay format they use as the original dataset.

We appreciate this observation, and have validated our predictions for two inhibitor combinations in the commercially available in vitro assay (Eurofins DiscoverX KdELECT) also used to generate the input dataset (Davis *et al.* 2011). These validation experiments (Figure 6; Figure 6 Figure Supplement 1) exhibit excellent matching between predicted and observed K_d_s. In most cases the accuracy of the K_d_ of combinations of compounds is near the accuracy limit set by individual compounds. Additionally, we edited the manuscript to point out that our MMS predictions are most accurate when the same assay format is used for the input data and the tested mixtures.

2) The paper is not well organized and not easily readable. The whole manuscript should be reorganized in a more concise and logical way.

We are grateful for the opportunity to make our work more readable and accessible to our audience, and have fundamentally restructured numerous parts of the work. This included cutting paragraphs of too domain-specific details from the introduction, making the explanation of the MMS method in the body of the paper more concise, and limiting conversation of predictions in the Results section to just the most important observations in addition to numerous other smaller changes. We hope that the reviewers appreciate the lengths to which we have gone to revise the work and make it more accessible to a general audience.

3) The authors should carefully explain and elaborate in the main text why the proposed methodology presents a real advantage over the most selective inhibitor.

We appreciate that this important question has been posed. We point out that the real advantage of the inhibitor combination is the both the improvement in selective inhibition for a single target and the ability to inhibit multiple kinase targets with optimal selectivity. Our new in vitro multitargeting experiments highlight these cases.

The apparent fold-selectivity for the target versus off-targets does change, and for the off-targets we validate in our new rational multitargeting experiments in this work, the observed selectivity advantage is considerable. Specifically, for the subset of off-targets studied in this work, the fold-selectivity advantage of Mixtures A and B over the most selective single inhibitors that were also potent against target kinases were 28.7-fold and 171-fold more selective, respectively (Figure 6 Figure Supplement 1).

We are careful to note that we only validated a subset of off-targets as opposed to the full set of hundreds of off-target human kinases, and that based upon the accuracy of our predictions, we would expect the full selectivity advantage of inhibitor Mixtures A and B tested in our work to look more like our predictions (Figure 6c) rather than what might be observed for any given subset of off-targets. In the case of Mixture A, this means reducing the off-target activity of potently inhibited off-target kinases. In the case of Mixture B, this means an average decrease in off-target activity of 7.5% across 385 unique off-target kinases. We emphasize that all off-targets tested in the in vitro assays are reported in this work, none were excluded from reporting.

4) The authors should conduct a comparative analysis of their methods with other similar methods to highlight the real advantage.

Thank you for this excellent suggestion. Other selectivity scoring metrics are designed to describe the selectivity of single compounds; our JSD metric is explicitly designed to score the cumulative activity of mixtures of compounds in addition to single compounds, and at user-defined levels of on-target activity. We provide direct comparisons of our JSD scoring metric with other representative metrics (Gini, a foldpotency target-specific relative selectivity term, and S-Scores) across three datasets (Figure 2 Figure Supplement 2).

While we understand the value of comparing new techniques to prior ones, we are not aware of prior techniques for quantifying the selectivity of mixtures of inhibitors or for predicting combinations of selective inhibitors. This novelty is part of the value of this work.

Reviewer #1 (Recommendations for the authors):The authors should assess at least one combination of kinase inhibitors in the same assay format they use as the original dataset. This is the only way to properly determine how well this method can work. The have been many reported differences between biochemical and cellular kinase assays and the formats should not be used interchangeably for this work.

We thank the reviewer for the excellent suggestions and observations. We have profiled two mixtures of inhibitors in the same assay as the original dataset (DiscoverX) and observe excellent matching between predicted and observed K_d_ values, and for the subset of off-targets tested, excellent selectivity (Figure 6e, Figure 6f). We are careful to note in our manuscript that we expect the whole-kinome selectivity advantage of the tested mixtures to be more similar to our predictions rather than the subset of off-targets tested.

The strength of this method relates to the application of kinase inhibitor cocktails to the study of biology. The discussion of polypharmacology should be removed. The authors cite that PK/PD would be needed to predict drug combinations in patients. Unless the authors are interested in looking at these factors, including CYP inhibition and induction to assess potential drug-drug interactions, this is too preliminary to include in the manuscript. Furthermore, there is no biological support for the polypharmacology claims – the authors would need to show that their proposed inhibitor combinations have a pharmacological benefit.

We have removed the bulk of our discussion on polypharmacology, and leave only limited mention in order to provide context for why multitargeting is valuable, and in the limitations section to indicate why the method should not be applied in certain settings. We have removed paragraphs from the introduction commenting on clinical toxicology. We emphasize the implementation of MMS in the study of biological systems, and not as a tool for proposing drug combinations. Additionally, we have even changed the title of our manuscript replacing polypharmacology with multitargeting. We hope that these changes reflect how seriously we take these concerns and our agreement with the reviewer’s assessments.

Reviewer #2 (Recommendations for the authors):Please, find below some more specific comments that, I hope, might help the authors improve the quality and clarity of their manuscript.The whole development of the method is based on the PKIS-2 set, however, it is not described in the manscript, beyond saying that it contains 645 inhibitors for 406 kinases.

We apologize for the confusion and we have added a new section to the Materials and methods (Dataset Sources) in the manuscript to describe the data sets used in this study.

Moreover, most of the data shown comes from simulations, where the authors model the "average" inhibitor in the PKIS-2, but I am sure that there is a wide heterogeneity, including kinases with many inhibitors and others with only one. Similarly, the range of on- and off-targets hit by each inhibitor must also be very heterogeneous. Without seeing a deep analysis of the PKIS-2 dataset, it is difficult to understand the validity of the simulations.

Thank you for raising this point. To clarify, the simulated datasets address only a side question: how much more useful would this method be if we had access to much larger inhibitor data sets? These are likely to exist in industry but are currently not openly available, however we expect that larger datasets will become available in the future. What the characteristics of these future datasets will be, we don’t know, but we take a reasonable guess based on the largest dataset available to us (PKIS2-645) to describe expected trends in MMS performance as a function of dataset size and compound selectivity. Importantly, for our purposes it was not a requirement to perfectly represent the selectivity of PKIS2-645 compounds since it is unknown what the profile of other compounds in other future datasets might be.

The average selectivity of the simulated inhibitors are quite similar but slightly greater than true PKIS2645 compounds, since they were based upon the average selectivity of PKIS2-645 compounds and the overall selectivity distribution for PKIS2-645 compounds is skewed. The simulated datasets reflect the heterogeneity in selectivity of the average compounds.

We have made an additional figure (Figure 3 Figure Supplement 1) to help clarify these details in regard to the similarity between simulated and true inhibitors (Figure 3 Figure Supplement 1a) and the selectivity of the simulated compounds from the parent selectivity profiles (Figure 3 Figure Supplement 1b).

Related to the point above, the applicability of the methods seems really low, with very few kinases being able to minimally benefit from the combination of inhibitors.

We appreciate this comment, and hope that our additional analyses, particularly those oriented towards multitargeting, help remove these concerns. We note in our work that selectivity improvement for single kinase targets is in most cases relatively modest, but that the predicted selectivity improvement for even pairs of kinases from the same datasets is much greater (Figure 6a). Our in vitro experiments emphasize the very substantial predicted (Figure 6c) and observed (Figure 6f) effect size for multitargeting applications.

The only validation of the strategy is presented in Figure 5, where the authors use nanoBRET in cells and cell lysates to quantify whether a combination of inhibitors could maintain the inhibition level on MAPK14 while reducing the off-targets binding. As shown, the most difficult cross-binding to avoid is with MAPK11 and, in this case, the superiority of the combination is very minor (if any). Indeed, the combination slightly decreases the binding with MAPK11, but it also goes down for the intended target.

Thank you for this observation, we agree that the improvement in selectivity is relatively small for this particular case. The combination was also less potent as expected.

The primary value of these data is that they illustrate the accuracy of MMS predictions inside cells, and they emphasize that for greatest accuracy predictions should be generated using single-compound target engagement data from the same system a combination would be deployed in. We have revised associated sections of our work to emphasize these points.

We have also performed the additional noted in vitro validation experiments to illustrate much larger effect sizes with respect to selectivity gains (Figure 6f, Figure 6 Figure Supplement 1).

I also find the text difficult to follow. In addition to the PKIS-2 data, the authors also play with other kinase-inhibitor sets, but they are not properly described and it is difficult to understand if they are being used as a support or to fine-tune the methodology.

Thank you, this feedback is very useful to us and identifies room for improvement in communication of our results. The datasets used in this work contain different inhibitors, targets, and one dataset (Klaeger *et al*) was generated with a different assay compared to the other three. All four of the datasets are of substantial quality and are frequently used for computational analyses of kinase-inhibitor interactions. We have added an additional section to our Materials and methods section (Dataset Sources) that describe these datasets in more detail and their particular value to this work.

Additionally, I also find some of the figures hard to interpret (i.e. off-target kinase density vs off-target activity). The concept is pretty simple, and the authors should find a clearer way to communicate their results.

We understand that the way we have communicated our data in some figures may not be immediately familiar among a broad audience, and that the term activity, which is central to this work, may not have been adequately described in our previous submission.

With the exception of a few relevant cases, we have removed plots with probability density functions from the majority of figures. We have added a section at the beginning of our Results labeled “Data Types and Definition” in order to help clearly communicate how to interpret and read these plots. We also add explicit definitions for terms like activity.

Reviewer #3 (Recommendations for the authors):A few other notices.1. The formats of "Figure " and "Figure" can be unified throughout the paper.

Thank you for this observation. We have unified all figure references.

2. Would it be possible to create a statistical table summarizing the subfigures from Figure 3-supplemental 1 to 6, as well as Figure 4-supplementals? This would facilitate the authors' understanding of the results and enable them to grasp the meaning.

Thank you, we appreciate this suggestion. Upon further reflection we acknowledge that these figures are confusing and have decided to remove them from the work since they are not required to convey the important findings of the associated experiments. The results for single-target analyses are summarized in our condensed Figure 4 Figure Supplement 1, and the set of critical results from our simulated analyses are reported in our updated Figure 3.

[Editors’ note: what follows is the authors’ response to the second round of review.]

The manuscript has been improved but there are some remaining issues that need to be addressed, as outlined below:One of the reviewers has raised concerns on the analysis and interpretation of the extra conducted results as below. The authors should carefully discuss the results obtained, in particular for the concept of "rational multi-targeting". One referee has recommended to split it out. Please amend the manuscript accordingly (introduction, discussion and conclusions, etc).

We have amended the introduction and discussion to clarify the benefits and limitations of the method applied to single kinases and multiple kinases.

We appreciate the reviewer’s suggestion for splitting the multi-targeting section out. However, we realized during the review process the challenges in efficiently communicating the technicalities of the study. We believe that the current manuscript links the different sections of the study in the most logical order:

1.) Development of a selectivity score and MMS combinations.

2.) Application of MMS to single kinases (as predicted with modest benefit).

3.) Application to multiple kinases where we can show the real benefit of MMS even with clinical relevance in the case of Abl/Fyn/Lyn. MMS can do something that single inhibitors cannot.

4.) Discussion of MMS strengths and limitations for single and multiple kinases / need for larger datasets.

We believe that multiple manuscripts consisting only of parts of the study would lack the overall impact of the combination of all parts within the current manuscript.

Reviewer #2 (Recommendations for the authors):I thank the authors for their answers to my comments. However, to be honest, I am a bit puzzled. The whole article was about the possibility of achieving a higher kinase specificity (and fewer off-target effects) with combinations of kinase inhibitors than with the most specific inhibitor for a given kinase.

We appreciate that the reviewer agrees that inhibitor combination is a good idea.

We would like to clarify that:

a) the combination of kinase inhibitors behave as predicted against single kinases (Figure 5 C,D,E and Figure 6E),

b) specificity is predicted to be improved by inhibitor combinations for up to 24 different kinases – this strongly depends on the available datasets for analysis (Figure 4) and on the dataset size (Figure 3D).

c) We have updated the introduction to mention the predicted and observed modest benefit of the method single kinase.

“As predicted, we find that the benefit of combinations depends dramatically on the available inhibitor dataset to inform combinations. While we predict and find a modest specificity improvement for single kinases, the benefit for multiple kinase inhibition is more pronounced.”

My main criticism (also highlighted by other Reviewers) is that it seems like a good idea, but the results do not support the hypothesis. Moreover, the applicability seems very limited to only a few kinases. Now, without really changing the bulk of the article, the authors have included a new figure (Figure 6) and it seems that the power of the MMS method is to identify drug combinations that are "more specific" in inhibiting combinations of kinases.

We would like to clarify that the concept of multi-targeting/polypharmacology and multiple kinase inhibition was part of the original submission (see title and old Figure 6).

However, obviously, the available inhibitors were never meant to do this function. Besides, the advantage of inhibiting two "random" kinases is very uncertain. As the paper is now, it seems like if the authors introduce a problem (i.e. low specificity of kinase inhibitors) that they cannot solve, and then change the question and find some examples where the methodology works. I am sorry not to be more positive but, in my opinion, the first version of the paper described a good idea that did not work. This second version reads like they are trying to conceal the results to sell a different success story. Again, in my opinion, it was a good idea worth testing, and it would be more honest to openly discuss the results obtained. Now this is a different paper, with a different focus, title, etc, not a resubmission. And should be treated as such.

We are grateful again for the reviewer’s appreciation of the idea and would like to point out that the method works as predicted for single and multiple kinases. We carefully point out the limitations of this approach and how larger datasets would increase its broader benefit (see introduction and discussion).

We appreciate the reviewer’s suggestion for splitting the multi-targeting section out. However, we realized during the review process the challenges in efficiently communicating the technicalities of the study. We believe that the current manuscript links the different sections of the study in the most logical order:

1.) Development of a selectivity score and MMS combinations.

2.) Application of MMS to single kinases (as predicted with modest benefit).

3.) Application to multiple kinases where we can show the real benefit of MMS even with clinical relevance in the case of Abl/Fyn/Lyn. MMS can do something that single inhibitors cannot.

4.) Discussion of MMS strengths and limitations for single and multiple kinases / need for larger datasets.

We believe that multiple manuscripts consisting only of parts of the study would lack the overall impact of the combination of all parts within the current manuscript.